JCB | Journal of Cell Biology

# V-ATPase is a universal regulator of LC3-associated phagocytosis and non-canonical autophagy

Kirsty M. Hooper[1], Elise Jacquin[1,2], Taoyingnan Li[3,4], Jonathan M. Goodwin[5], John H. Brumell[3,4,6,7], Joanne Durgan[1], and Oliver Florey[1]

Non-canonical autophagy is a key cellular pathway in immunity, cancer, and neurodegeneration, characterized by conjugation of ATG8 to endolysosomal single membranes (CASM). CASM is activated by engulfment (endocytosis, phagocytosis), agonists (STING, TRPML1), and infection (influenza), dependent on K490 in the ATG16L1 WD40-domain. However, factors associated with non-canonical ATG16L1 recruitment and CASM induction remain unknown. Here, using pharmacological inhibitors, we investigate a role for V-ATPase during non-canonical autophagy. We report that increased V0–V1 engagement is associated with, and sufficient for, CASM activation. Upon V0–V1 binding, V-ATPase recruits ATG16L1, via K490, during LC3-associated phagocytosis (LAP), STING- and drug-induced CASM, indicating a common mechanism. Furthermore, during LAP, key molecular players, including NADPH oxidase/ROS, converge on V-ATPase. Finally, we show that LAP is sensitive to *Salmonella* SopF, which disrupts the V-ATPase–ATG16L1 axis and provide evidence that CASM contributes to the *Salmonella* host response. Together, these data identify V-ATPase as a universal regulator of CASM and indicate that SopF evolved in part to evade non-canonical autophagy.

## Introduction

Autophagy is a fundamental degradative process required for nutrient recycling, clearance of damaged organelles, and pathogen responses. During autophagy, a collection of ATG proteins induce the formation and maturation of autophagosomes, which enwrap target cargo destined for lysosomal degradation (Choi et al., 2013). A subset of ATG proteins also acts in parallel to mediate "non-canonical autophagy," a pathway targeting single-membrane, endolysosomal compartments instead (Florey and Overholtzer, 2012; Heckmann et al., 2017). A diverse set of stimuli and cellular processes induce non-canonical autophagy, including stimulation of the TRPML1 calcium channel (Goodwin et al., 2021), activation of the stimulator of interferon genes (STING) pathway (Fischer et al., 2020), and disruption of endolysosomal ion balance by pharmacological agents, such as *Helicobacter pylori* VacA toxin, or infection with influenza A virus (Fletcher et al., 2018; Florey et al., 2015; Jacquin et al., 2017). Non-canonical autophagy is also associated with endocytic and engulfment processes, including entosis (Florey et al., 2011), LC3-associated endocytosis (Heckmann et al., 2019), and LC3-associated phagocytosis (LAP; Martinez et al., 2015; Sanjuan et al., 2007). It has also displayed essential functions in the killing and clearance of pathogens (Gluschko et al., 2018; Hubber et al., 2017; Martinez et al., 2015), antigen presentation (Fletcher et al., 2018; Ma et al., 2012; Romao et al., 2013), clearance of and cytokine responses to apoptotic cells (Cunha et al., 2018; Florey et al., 2011; Martinez et al., 2011; Martinez et al., 2016; Martinez et al., 2015), clearance of accumulated β-amyloid (Heckmann et al., 2020; Heckmann et al., 2019), vision (Kim et al., 2013), response to influenza infection (Fletcher et al., 2018; Wang et al., 2021), and lysosome biogenesis (Goodwin et al., 2021). While the functional importance of non-canonical autophagy is clear, the mechanisms underlying the regulation and function of this pathway remain incompletely understood.

The canonical and non-canonical autophagy pathways share overlapping machineries but harbor key differences. The non-canonical pathway is independent of the upstream autophagy regulators ATG9, mTOR, and the ULK1 initiation complex (ULK1/2, FIP200, ATG13, and ATG101; Florey et al., 2011; Martinez et al., 2015). However, both pathways harness the core ATG8 lipidation machinery (ATG3, 4, 5, 7, 10, 12, 16L1). During canonical autophagy, ATG8s are conjugated exclusively to phosphatidylethanolamine (PE) at double-membrane autophagosomes (Ichimura et al., 2000), where they support cargo selection, membrane elongation, and phagophore closure (Johansen and Lamark, 2020). In non-canonical autophagy, ATG8s are conjugated instead to single membrane,

[1]Signalling Programme, Babraham Institute, Cambridge, UK;   [2]Institut national de la santé et de la recherche médicale UMR-S 1193, Université Paris-Saclay, Châtenay-Malabry, France;   [3]Department of Molecular Genetics, University of Toronto, Toronto, Ontario, Canada;   [4]Cell Biology Program, Hospital for Sick Children, Toronto, Ontario, Canada;   [5]Casma Therapeutics, Cambridge, MA;   [6]Institute of Medical Science, University of Toronto, Toronto, Ontario, Canada;   [7]SickKids Inflammatory Bowel Disease Centre, Hospital for Sick Children, Toronto, Ontario, Canada.

Correspondence to Oliver Florey: oliver.florey@babraham.ac.uk.

endolysosomal compartments, such as phagosomes, endosomes, or entotic vacuoles, and this process is termed conjugation of ATG8 to single membranes (CASM) and involves alternative conjugation to both phosphatidylserine (PS) and PE (Durgan et al., 2021).

ATG16L1 acts as a critical molecular hub, directing canonical and non-canonical autophagy at different sites via different domains. During canonical autophagy, ATG16L1 is recruited to form autophagosomes through its coiled-coil domain, which interacts with WIPI2 and FIP200, thereby specifying the site of ATG8 conjugation (Dooley et al., 2014; Gammoh et al., 2013). During non-canonical autophagy, ATG16L1 is recruited instead to preformed endolysosomal membranes to drive CASM. The precise molecular mechanisms underlying this alternative recruitment remain unclear, but the ATG16L1 WD40 C-terminal domain (CTD) and certain ATG16L1 lipid-binding motifs are indispensable (Fletcher et al., 2018; Lystad et al., 2019; Rai et al., 2019). A single point mutation in ATG16L1, K490A, renders cells competent for canonical autophagy, but deficient in non-canonical autophagy, providing a highly specific genetic system to dissect these closely related pathways (Fletcher et al., 2018; Goodwin et al., 2021; Lystad et al., 2019).

The molecular mechanisms of non-canonical autophagy are most commonly studied in the context of LAP. LAP is dependent on Rubicon/Vps34, which mediates PI3P formation, and NADPH oxidase, which drives the generation of reactive oxygen species (ROS; Huang et al., 2009; Martinez et al., 2015). However, the mechanisms by which ROS generation activates the LAP pathway remains a key, unanswered question in the field. In addition, while LAP is dependent on Rubicon and ROS, other non-canonical autophagy processes, such as entosis and drug-induced CASM, remain independent (De Faveri et al., 2020; Florey et al., 2015; Jacquin et al., 2017), suggesting these molecular players are stimulus-specific inputs, rather than being universally required for CASM. This raises another important, open question regarding a molecular mechanism that might unify the diverse forms of non-canonical autophagy.

In considering a possible universal regulator for non-canonical autophagy, we turned our attention to the vacuolar-type ATPase (V-ATPase) as a candidate. V-ATPase is a multi-subunit protein complex that acts as a molecular pump, generating proton gradients within intracellular compartments. It is required to acidify lysosomes and support the degradation of autophagosome inner membranes and their engulfed material. As such, V-ATPase is generally thought to play a terminal, degradative role in autophagy-related processes. However, in the context of non-canonical autophagy, we hypothesized that V-ATPase might also play an upstream, activating role based on several lines of evidence. Firstly, inhibition of V-ATPase by Bafilomycin A1 (BafA1) blocks CASM induced by LAP, entosis, ionophore, and lysosomotropic drug stimulation, TRPML1 activation, and STING agonists (Fischer et al., 2020; Florey et al., 2015; Goodwin et al., 2021; Jacquin et al., 2017), indicating a broad but undefined role in non-canonical autophagy. Secondly, a direct interaction has recently been identified between V-ATPase and ATG16L1 during *Salmonella* infection, which can be inhibited by the effector protein SopF (Xu et al., 2019). While this interaction has been defined in the context of xenophagy, it is striking that it depends on the ATG16L1 WD40 CTD, a domain specifically required for non-canonical autophagy, thus hinting at a possible connection to CASM. Finally, and in line with this reasoning, STING activation induces non-canonical autophagy in a SopF-sensitive fashion in a process thus termed V-ATPase–ATG16L1 induced LC3B lipidation (VAIL; Fischer et al., 2020). Integrating these diverse observations, we hypothesized that V-ATPase may represent a universal regulator of non-canonical autophagy.

In this study, we investigate the role of V-ATPase in CASM, identifying a requirement for enhanced V0–V1 association and direct ATG16L1 engagement across a wide range of non-canonical autophagy processes. We also define the interrelationships between V-ATPase, NADPH oxidase, and ROS during LAP, providing new mechanistic insight into this specific process. Finally, we explore the roles of V-ATPase and non-canonical autophagy during *Salmonella* infection, concluding that CASM represents a host-pathogen response, evaded by the SopF effector.

## Results

### V-ATPase is recruited to phagosomes prior to LC3 lipidation

During LAP, the fusion of phagosomes with late endosomes/lysosomes and their associated acidification via V-ATPase is assumed to occur after the recruitment of LC3 as a terminal step (Martinez et al., 2015). However, inhibition of V-ATPase with BafA1 blocked LC3 recruitment during LAP and other non-canonical autophagy processes, suggesting V-ATPase may play an additional, upstream role (Florey et al., 2015). To investigate this notion further, we revisited the dynamics of LAP using live imaging.

First, fluorescent-tagged markers of late endosomes/lysosomes were monitored in relation to GFP-hLC3A (human LC3A) during phagocytosis of opsonized zymosan particles (LAP). Strikingly, in all phagosomes analyzed, we observed early recruitment of the late endosome marker, RFP-Rab7, to newly formed phagosomes, occurring prior to GFP-hLC3A recruitment (Fig. 1 a and Video 1). Similarly, the lysosome marker, LAMP1-RFP, was also recruited to all phagosomes analyzed before GFP-hLC3A (Fig. 1 b and Video 2). By measuring the duration between onset of markers, we confirmed that phagosomes matured through a Rab7–LAMP1–LC3 sequence (Fig. 1 c). The same maturation profile was also observed during entosis, another macroscale engulfment process that utilizes non-canonical autophagy (Fig. S1). Together, these data imply that late endosomes/lysosomes fuse with the phagosome prior to the initiation of non-canonical autophagy signaling.

We next sought to visualize V-ATPase directly. This large, multi-subunit protein complex can be challenging to monitor in live cells as fluorescent tags may interfere with function. Therefore, to assess V-ATPase and LC3 recruitment to phagosomes, immunofluorescence staining was performed at specific time points on fixed cells. At 20 min post-zymosan incubation, the majority of phagosomes were positive for both the V1A

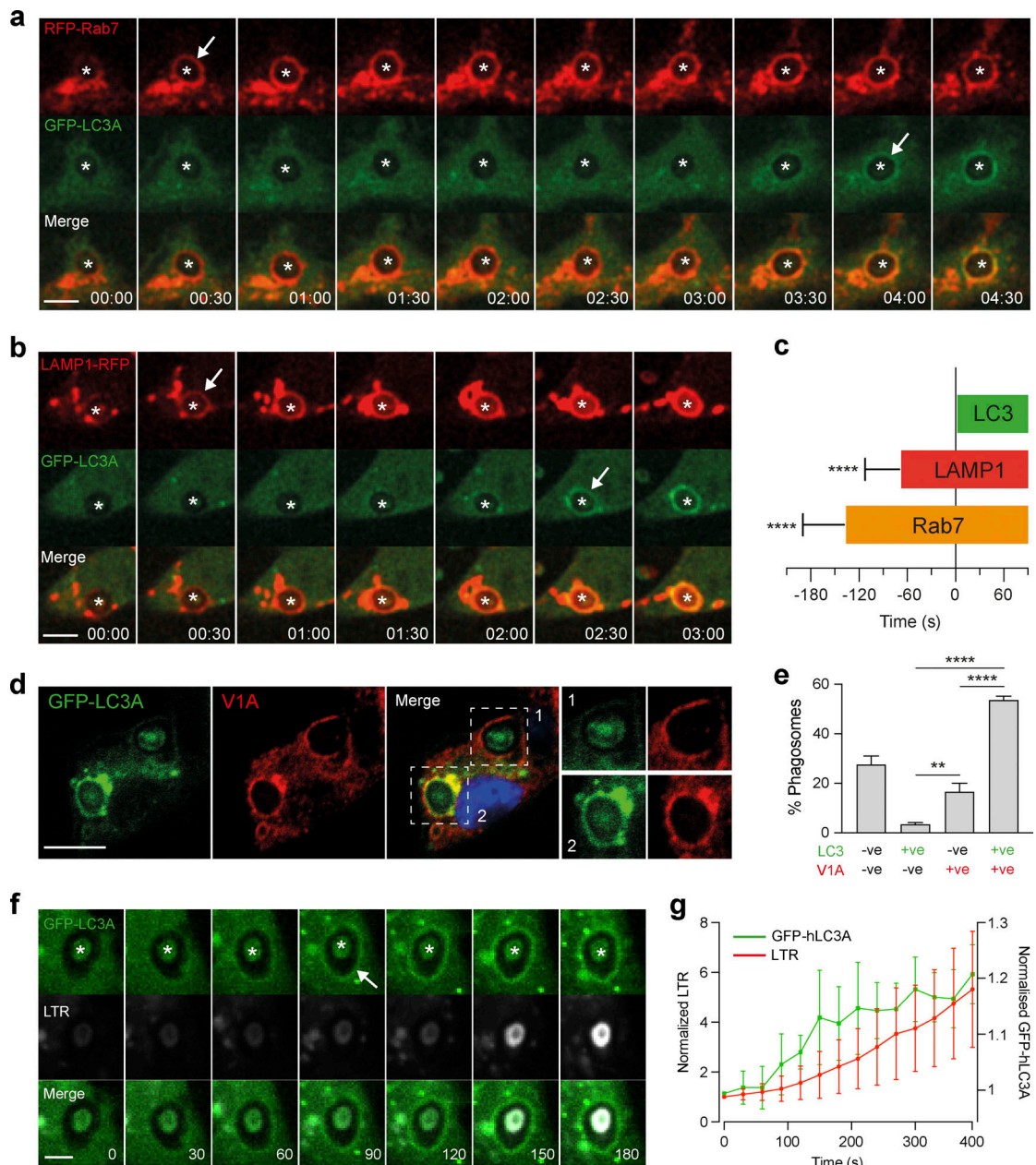

Figure 1. **Rab7, LAMP1, and V-ATPase recruit prior to LC3 during LAP. (a and b)** Representative time-lapse confocal microscopy images of OPZ phagocytosis in RAW264.7 cells expressing GFP-hLC3A and RFP-Rab7 (a) or LAMP1-RFP (b). Asterisks denote phagosomes and arrows mark acquisition of fluorescent markers. Scale bar, 5 µm, cropped box, 5 µm; time, min:s. **(c)** Quantification of fluorescent marker acquisition from 10 phagosomes in relation to GFP-hLC3A (time 0). Data represent mean ± SD. ****, P < 0.0001, unpaired t test. **(d)** Confocal images of OPZ containing phagosomes after 20 min, stained for GFP-hLC3A and ATP6V1A. Scale bar, 5 µm. **(e)** Quantification of phagosome markers from d. Data represent mean ± SEM from three independent experiments with >100 phagosomes analyzed per experiment. ****, P < 0.0001; **, P < 0.001, one-way ANOVA followed by Tukey multiple comparison test. **(f)** Representative time-lapse confocal microscopy images of LTR and GFP-hLC3A during OPZ phagocytosis. Asterisks denotes phagosomes and arrow marks acquisition of GFP-hLC3A. Scale bar, 2 µm; time, s. **(g)** Quantification of phagosome markers from f. Data represent mean ± SD from eight phagosomes.

subunit of V-ATPase and GFP-hLC3A (Fig. 1, d and e). Importantly, significantly more V1A positive, GFP-hLC3A negative phagosomes were detected than V1A negative, GFP-hLC3A positive (Fig. 1, d and e), suggesting that V1A is likely to be recruited first. We can exclude the possibility that GFP-hLC3A had transiently translocated before fixation as GFP-hLC3A clearly persisted on phagosomes for longer than 20 min in live imaging studies (Durgan et al., 2021). Therefore, our data indicate that,

like Rab7 and LAMP1, V-ATPase is present on phagosomes prior to LC3. These findings are consistent with a previous study indicating that V-ATPase recruits to phagosomes at an early stage of their maturation (Sun-Wada et al., 2009).

Finally, we used LysoTracker Red (LTR) to monitor phagosome acidification in relation to GFP-hLC3A recruitment. Our analyses showed that GFP-hLC3A recruited to phagosomes prior to significant acidification (Fig. 1, f and g). These data suggest

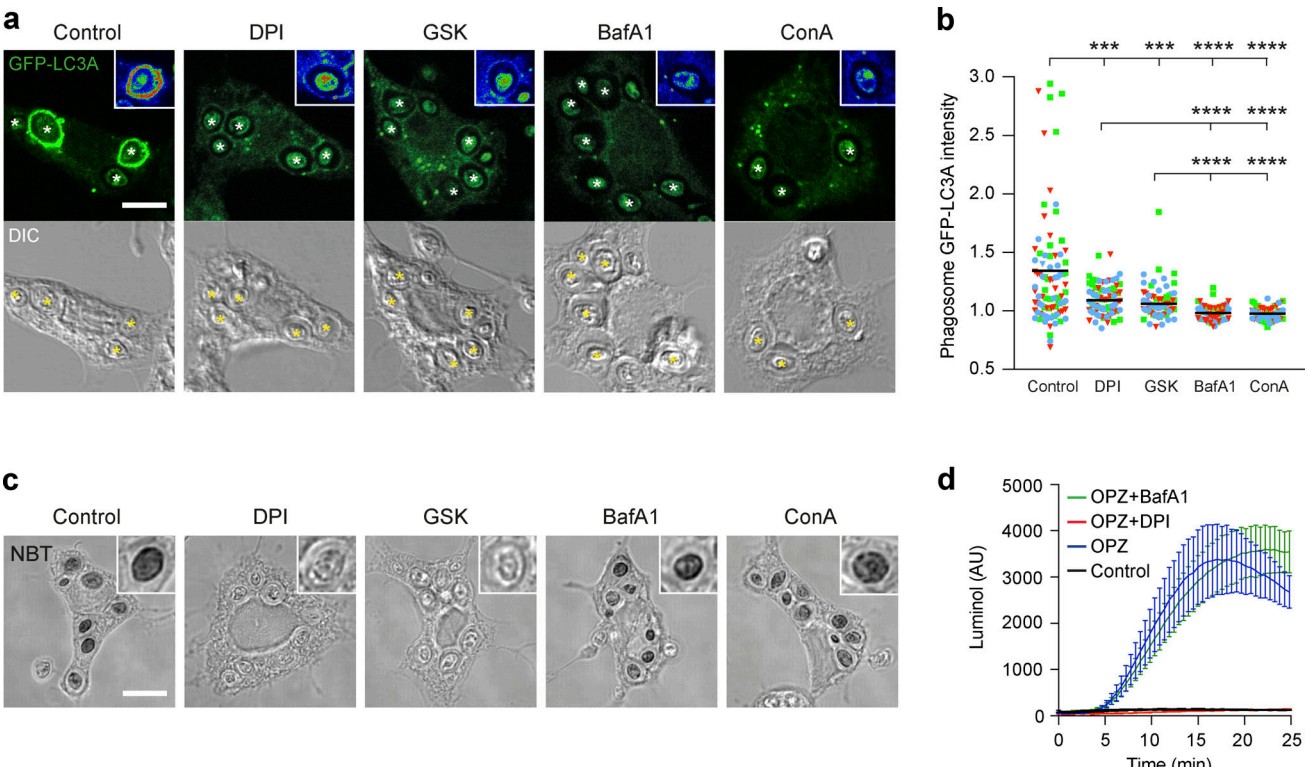

**Figure 2. NADPH oxidase and V-ATPase are both required for LAP. (a)** Confocal DIC and GFP-hLC3A images of OPZ phagocytosis in RAW264.7 cells treated with DPI (5 µM), GSK (10 µM), BafA1 (100 nM), or ConA (100 nM). Cropped insert showing pseudo-colored GFP fluorescence intensity. Asterisks denote phagosomes. Scale bar, 5 µm, cropped boxes, 7 µm. **(b)** Quantification of GFP-hLC3A intensity at phagosomes in cells treated as above. Data represent the mean of individual phagosome measurements from three independent experiments (red triangle, green square, blue circle); ****, P < 0.0001; ***, P < 0.0007, one-way ANOVA followed by Dunnett's T3 multiple comparison test. **(c)** Representative confocal DIC images of NBT/formazan deposits in phagosomes from control, DPI, GSK, BafA1, and ConA treated RAW264.7 cells. Inserts are zoomed in phagosomes. Scale bar, 5 µm, cropped boxes 7 µm. **(d)** Luminol measurements of ROS during OPZ phagocytosis in RAW264.7 cells treated with DPI or BafA1. Data represent the mean ± SEM from three independent experiments. AU, arbitrary unit.

that while V-ATPase is physically present on the phagosome prior to LC3 recruitment, acidification of the compartment occurs later and is not a pre-requisite step.

Together, these data indicate that components of the late endosome/lysosome network, including V-ATPase, are recruited to phagosomes before the onset of LC3 lipidation during LAP, but are not yet successful in acidification. These data question the view that V-ATPase performs a terminal role and are consistent with a possible upstream, regulatory function that may be pump independent.

**NADPH oxidase and V-ATPase activities are both required for LAP**

A major molecular mechanism implicated in LAP involves NADPH oxidase-mediated ROS generation (Martinez et al., 2015); however, the specific mechanistic role of ROS remains unclear. We next considered the possibility of a relationship between NADPH oxidase and V-ATPase during LAP. Using the well-characterized pharmacological inhibitors diphenyleneiodonium (DPI) and GSK2795039 (GSK) to inhibit NADPH oxidase, and BafA1 and Concanamycin A (ConA) to inhibit V-ATPase, we observed a reduction in GFP-hLC3A recruitment to phagosomes with all (Fig. 2, a and b). Closer inspection of phagosomal

fluorescence intensities suggests that BafA1 and ConA yielded more pronounced inhibition compared with NADPH oxidase inhibition (Fig. 2 a). These data confirm that both NADPH oxidase and V-ATPase are required for LAP. A limitation of this data is the reliance on pharmacological inhibition of V-ATPase and NOX2. Therefore, in the future, it would be of interest to genetically perturb these complexes.

Next, the effect of each inhibitor on phagosomal ROS generation was analyzed, using Nitroblue Tetrazolium (NBT; Fig. 2 c) and luminol based assays (Fig. 2 d). As shown in Fig. 2 c, $O_2^-$ mediated insoluble formazan deposition, in zymosan containing phagosomes, was abolished by DPI and GSK, but not by BafA1 or ConA treatment. In agreement with this, DPI treatment blocked zymosan induced ROS production, while BafA1 had no effect (Fig. 2 d). Together, these data establish that while NADPH oxidase is required for ROS generation in phagosomes, V-ATPase activity is not. Moreover, these results show that V-ATPase inhibition blocked LAP, even in the presence of a ROS burst. As such, we can deduce that NADPH oxidase-mediated ROS generation is not sufficient to induce LAP, and that V-ATPase acts either downstream or in parallel to promote LC3 lipidation to the phagosome membrane.

## ROS modulates phagosome acidification to drive LAP

To test whether NADPH oxidase might function directly upstream of V-ATPase during LAP, the relationship between ROS and phagosome acidification was assessed using Lysotracker and live-cell confocal microscopy in the presence or absence of DPI. Strikingly, inhibition of NADPH oxidase significantly increased Lysotracker intensity, signifying increased acidification of zymosan containing phagosomes by V-ATPase (Figs. 3 a and S2). These data agree with previous studies (Mantegazza et al., 2008; Savina et al., 2006), which proposed that production of phagosomal ROS can constrain acidification by increasing membrane permeability or through the consumption of protons (H$^+$; Westman and Grinstein, 2020). According to this model, in the absence of ROS, more protons can accumulate, and thus the phagosome is more readily acidified.

To explore this further, we utilized chloroquine (CQ), a weak-base amine, which becomes protonated and accumulates in acidifying compartments, thereby "trapping" H$^+$ ions. We reasoned that CQ could artificially sequester H$^+$ in DPI-treated cells, providing an alternative way to limit protons in the absence of ROS. In line with our predictions, the addition of CQ countered the DPI-mediated increase in Lysotracker staining (Fig. 3, b and c). Importantly, CQ treatment also rescued GFP-hLC3A recruitment in the presence of DPI, connecting ROS generation and proton concentration directly to LAP (Fig. 3, b and d). To exclude the possibility that CQ rescued GFP-hLC3A recruitment by blocking cargo degradation, DPI-treated cells were co-treated instead with E-64d + pepstatin (cysteine protease and cathepsin inhibitors). Importantly, E-64d + pepstatin did not reverse the DPI-mediated inhibition of LAP (Fig. S3). Together, these data establish a relationship whereby high phagosomal ROS levels raise pH and drive LAP.

To interrogate this relationship in the absence of pharmacological manipulation, bone marrow–derived murine macrophage (BMDM) and dendritic cells (BMDCs) were compared. Previous studies have established that mouse BMDC phagosomes generate higher ROS levels and have a more neutral pH than BMDM phagosomes (Mantegazza et al., 2008; Savina et al., 2006). Using the luminol assay to measure ROS and Lysotracker staining to monitor acidification, we confirmed these observations (Fig. 3, e–g). Strikingly, we also found that BMDCs supported higher levels of LAP than BMDMs (Fig. 3, h and i), consistent with our model.

Thus, using both pharmacological and cell type-based manipulations, our data indicate that increased LAP is associated with higher ROS production and pH. These findings provide new insight into the molecular mechanisms underlying LAP and uncover a clear link between NADPH oxidase and V-ATPase activities.

## ROS modulates V-ATPase subunit recruitment

We next sought to understand how ROS alters V-ATPase function and how this results in the lipidation of LC3 to phagosomes. While modulation of phagosome pH does occur with ROS, our data do not support this as a direct mechanism. Firstly, while DPI and BafA1 both inhibited LAP, they yielded different effects on phagosomal pH (Fig. S2). Secondly, while BafA1 blocked LAP,

monensin, and many other drugs that raise lysosomal pH instead activate non-canonical autophagy (Jacquin et al., 2017). We, therefore, considered whether other, pump-independent features of V-ATPase might connect ROS and LAP, focussing first on V-ATPase subunit localization and recruitment.

During LAP, we observed robust recruitment of ATP6V1A, a subunit of the V-ATPase V1 sector, to zymosan containing phagosomes (Fig. 4 a). Notably, both DPI and BafA1 treatment reduced this ATP6V1A signal intensity (Fig. 4, a and b). This result may at first appear somewhat counter intuitive given that DPI increased the acidification of phagosomes (Fig. 3 a). However, we speculate that in the absence of H$^+$ consumption by ROS, the phagosome requires less V-ATPase activity to support acidification, and therefore recruits less ATP6V1A. In support of this interpretation, artificial sequestration of H$^+$ by CQ reversed the DPI-induced reduction in ATP6V1A staining (Fig. 4 c). To extend these observations, the recruitment of ATP6V1A to phagosomes was also compared in BMDCs and BMDMs. Consistent with the above data, BMDC phagosomes exhibited greater ATP6V1A recruitment than BMDMs (Fig. 4, d and e). Furthermore, inhibition of ROS production in BMDCs with DPI reduced V1A recruitment to their phagosomes accordingly, rendering them more similar to BMDMs (Fig. 4 e).

Taken together, these findings indicate that LAP activation correlates with V-ATPase V1 sector recruitment to the phagosome, which can be regulated by NADPH oxidase activity.

## Increased V0–V1 association drives non-canonical autophagy

V-ATPase activity can be regulated by the reversible association of the cytosolic V1 sector with the membrane-bound V0 sector to form a functional holoenzyme, a requisite for pump activity (Collins and Forgac, 2020). To address whether regulated V0–V1 association might play a role in non-canonical autophagy, two structurally distinct V-ATPase inhibitors, BafA1 and Saliphenylhalamide (SaliP), were exploited. These inhibitors bind to distinct sites on V-ATPase, both blocking pump activity (Xie et al., 2004) and raising lysosomal pH, as seen using LTR staining (Fig. 5, a and b). Importantly, however, while BafA1 acts to dissociate V0–V1, SaliP instead drives their association through a covalent adduct formation (Fig. 5 a; Kissing et al., 2015; Xie et al., 2004). As such, we can use these inhibitors to differentially manipulate V0–V1 association.

Treatment of WT MCF10A cells with BafA1 or SaliP led to an increase in LC3 lipidation (Fig. 5 c), which would typically be attributed to the inhibition of autophagic flux (Yamamoto et al., 1998). Consistent with this, in ATG13KO cells, which are deficient in canonical autophagy, BafA1 yielded no such increase in LC3 lipidation. Strikingly, however, SaliP still induced robust GFP-hLC3A lipidation co-localized with LAMP1 in the autophagosome-deficient ATG13 KO cells (Fig. 5, c and d), indicating that SaliP must specifically drive an alternative form of LC3 lipidation.

To explore whether SaliP-induced GFP-hLC3A lipidation was associated with CASM, we monitored GFP-hLC3A recruitment to single-membrane entotic corpse vacuoles (ECVs) as a well-established model for non-canonical autophagy, which is amenable to live imaging (Florey et al., 2015). Similar to monensin,

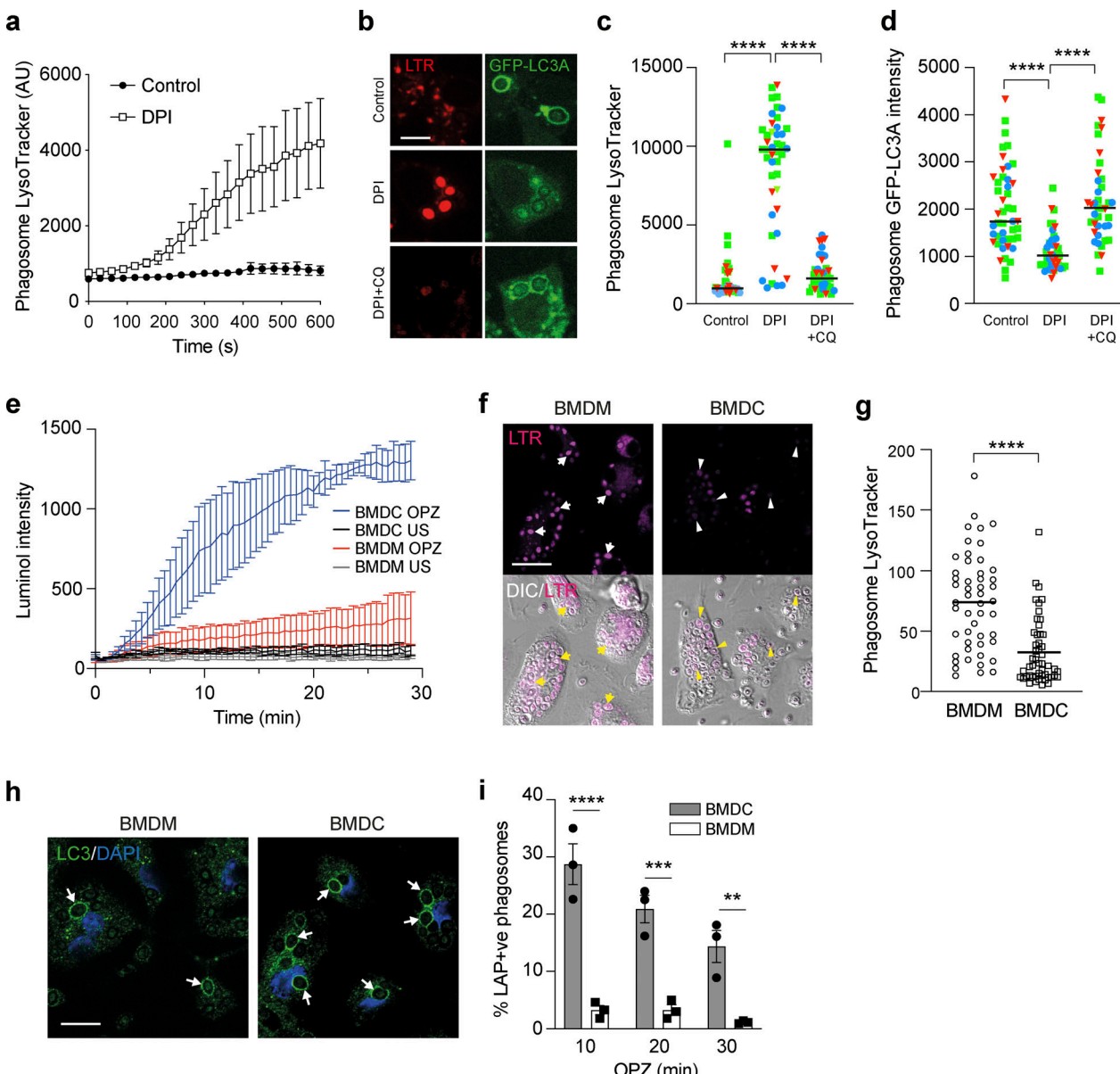

Figure 3. **LAP induction involves elevated ROS and raised pH in phagosomes. (a)** Confocal microscopy measurement of phagosome LysoTracker intensity over time in RAW264.7 cells stimulated with OPZ ± DPI (5 µM) pretreatment. Data represent mean ± SEM of nine individual phagosomes across multiple independent experiments. **(b)** RAW264.7 cells were pretreated with DPI (5 µM) or DPI + CQ (100 µM) prior to stimulation with OPZ for 25 min. Representative confocal images of LysoTracker and GFP-hLC3A are shown. Scale bar, 5 µm. **(c and d)** Quantification of Lysotracker (c) and GFP-hLC3A (d) at phagosomes in cells treated as in b. Data represent the mean of individual phagosome measurements from three independent experiments (red triangle, green square, blue circle). ****, P < 0.0001, one-way ANOVA followed by Tukey multiple comparison test. **(e)** Luminol measurements of ROS during OPZ phagocytosis in primary murine BMDC and BMDM cells. Data represent the mean ± SEM from three independent experiments. **(f)** Representative confocal images of LysoTracker staining and DIC in BMDC and BMDM cells after stimulation with OPZ for 25 min. Arrows denote phagosomes in BMDMs and arrowheads denote phagosomes in BMDCs. Scale bar, 20 µm. **(g)** Quantification of phagosome Lysotracker intensity in cells treated as in f. Data represent the mean of 50 individual phagosomes. ****, P < 0.0001, unpaired t test. **(h)** Representative confocal images of BMDC and BMDM cells stimulated with OPZ for 25 min and stained for LC3. Arrows denote LC3 positive phagosomes. Scale bar, 10 µm. **(i)** Quantification of LC3 positive phagosomes from cells stimulated with OPZ for the indicated times. Data represent mean ± SEM from three independent experiments. ***, P < 0.0001; ***, P < 0.0003; **, P < 0.003, two-way ANOVA with multiple comparisons. AU, arbitrary unit.

SaliP induced robust recruitment of GFP-hLC3A to ECVs, while BafA1 had no effect (Fig. 5, e and f). Furthermore, while BafA1 blocked monensin-induced GFP-hLC3A recruitment, SaliP had no such effect (Fig. 5 f). These data indicate that SaliP and BafA1 have opposing effects on drug-induced CASM, as observed on ECV membranes as a model system.

To extend these findings, we also analyzed physiological examples of entosis, in which transient recruitment of LC3 can be observed just prior to inner cell killing (Florey et al., 2011), in a manner inhibited by BafA1 (Fig. 5 g; Florey et al., 2015). Again, we observed a contrast between these V-ATPase inhibitors since SaliP had no effect on the frequency of GFP-hLC3A recruitment

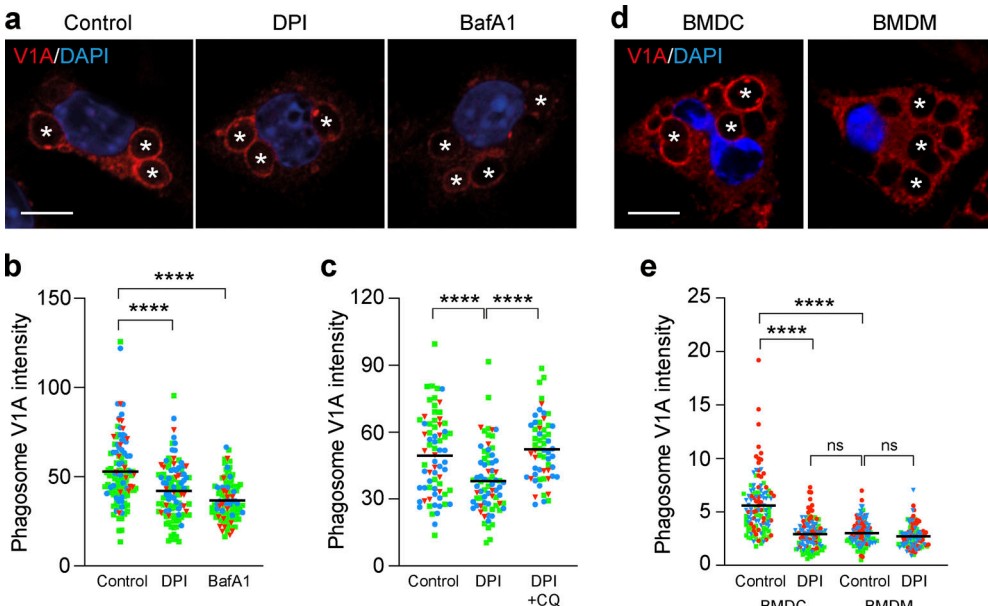

Figure 4. **Modulation of ROS regulates phagosomal V-ATPase recruitment during LAP. (a)** Representative confocal images of ATP6V1A following 25 min OPZ stimulation of RAW264.7 cells pretreated with either DPI (5 µM) or BafA1 (100 nM). Asterisks denote phagosomes. Scale bar, 5 µm. **(b)** Quantification of ATP6V1A phagosome intensity from cells treated as in a. Data represent the mean of individual phagosome measurements from three independent experiments (red triangle, green square, blue circle). ****, P < 0.0001, one-way ANOVA followed by Tukey multiple comparison test. **(c)** Quantification of ATP6V1A phagosome intensity from RAW264.7 cells pretreated with DPI or DPI + CQ (100 µM). Data represent the mean of individual phagosome measurements from three independent experiments (red triangle, green square, blue circle). ****, P < 0.0001, one-way ANOVA followed by Tukey multiple comparison test. **(d)** Representative confocal images of ATP6V1A following 25 min OPZ stimulation of BMDC or BMDM cells. Asterisks denote phagosomes. Scale bar, 5 µm. **(e)** Quantification of ATP6V1A phagosome intensity from BMDC and BMDM cells pretreated ± DPI (5 µM). Data represent the mean of individual phagosome measurements from three independent experiments (red triangle, green square, blue circle). ****, P < 0.0001, one-way ANOVA followed by Tukey multiple comparison test.

to entotic vacuoles (Fig. 5 g). Instead, SaliP significantly extended the duration of GFP-hLC3A at entotic vacuoles (Fig. 5 h and i; and Video 3).

Taken together, these findings indicate that while BafA1 inhibits CASM, SaliP instead induces GFP-hLC3A lipidation through CASM. These data suggest that forced V0–V1 association may be sufficient to trigger non-canonical autophagy.

To investigate this notion further, V0–V1 association was monitored in response to known inducers of CASM using established membrane fractionation assays (McGuire and Forgac, 2018; Stransky and Forgac, 2015). Strikingly, both monensin and CQ, two distinct pharmacological agents which disrupt lysosomal ion balance to activate non-canonical autophagy, promoted increased V0–V1 association (Fig. 6, a and b). Furthermore, a TRPML1 agonist (C8), which also drives CASM (Goodwin et al., 2021), similarly enhanced V0–V1 binding (Fig. 6, c and d). Notably, however, activation of canonical autophagy via mTOR inhibition (AZD8055) did not increase V0–V1 engagement, suggesting this mechanism is specific to the non-canonical pathway.

Together, these data indicate that enhanced V0–V1 engagement is a conserved feature of non-canonical, but not canonical, autophagy processes. Based on the use of SaliP, we propose that forced association of V0–V1 is sufficient for CASM activation and independent of V-ATPase pump activity. We conclude that non-canonical autophagy depends instead upon physical subunit

engagement, likely associated with structural changes within the V-ATPase complex itself.

**V0–V1 association drives V-ATPase–ATG16L1 interaction**
A direct interaction between V-ATPase and ATG16L1 was recently identified in the context of xenophagy (Xu et al., 2019). Intriguingly, this interaction involves the WD40 CTD of ATG16L1, a domain also essential for CASM (Fletcher et al., 2018). We thus hypothesized that V-ATPase, and specifically V0–V1 engagement, may drive CASM via comparable recruitment of ATG16L1. To test this, co-immunoprecipitation experiments were performed in cells expressing either FLAG-tagged WT ATG16L1 or a WD40 CTD point mutant (K490A), which is specifically deficient in non-canonical autophagy (Durgan et al., 2021; Fletcher et al., 2018). Under resting conditions, ATG16L1 did not interact with V-ATPase. However, during LAP, we observed robust interaction between WT ATG16L1 and the V-ATPase ATP6V1A and ATP6V1B2 subunits (Fig. 7, a and e; and Fig. S4, c and e). Importantly, this interaction was completely abolished by the K490A mutation, consistent with the molecular dependencies of non-canonical autophagy, while WT and K490A ATG16L1 were both able to pull down ATG5. This K490-dependent interaction between ATG16L1 and ATP6V1A was similarly induced by diverse non-canonical autophagy activators, including the STING ligand DMXAA (Figs. 7, b and e, and S4 c), TRPML1 activation with C8 (Figs. 7, c and f, and S4 c), monensin (Fig. S4, a–c),

Figure 5. **SaliP induces non-canonical autophagy and V-ATPase V0–V1 engagement. (a)** Cartoon showing the differential effects of BafA1 and SaliP on V0–V1 association. **(b)** Confocal images of LTR staining in WT MCF10A cells treated with BafA1 (100 nM) or SaliP (2.5 µM). Scale bar, 15 µm. Quantification of LTR intensity. Data represent mean ± SD from eight fields of view. ****, P < 0.0001. **(c)** WT and *ATG13⁻/⁻* MCF10A cells were treated with BafA1 (100 nM) or SaliP (2.5 µM) for 1 h. Western blotting was performed to probe for LC3 (I and II forms marked) and GAPDH. Quantification of LC3II/I levels. Data represent mean ± SEM from three independent experiments. ***, P < 0.0001; *, P < 0.02, unpaired *t* test. **(d)** Representative confocal images of *ATG13⁻/⁻* MCF10A cells stained for LAMP1 and GFP-hLC3A following treatment with SaliP (2.5 µM) for 1 h. **(e)** Confocal images of entotic corpse vacuoles in WT MCF10A cells treated with monensin (100 µM) for 1 h and stained for LAMP1 and GFP-hLC3A. Scale bar, 10 µm. **(f)** Quantification of GFP-hLC3A recruitment to LAMP1 positive entotic corpse vacuoles (ECVs) following treatment with monensin (Mon, 100 µM), SaliP (2.5 µM), or BafA1 (100 nM) for 1 h. Data represent mean ± SEM from three independent experiments. ****, P < 0.0001; **, P < 0.008, unpaired *t* test. **(g)** Quantification of GFP-hLC3A recruitment to entotic vacuoles during inner cell death in WT MCF10A cells treated ± BafA1 (100 nM) or SaliP (2.5 µM). Data represent mean ± SEM from three independent experiments. ****, P < 0.0001, unpaired *t* test. **(h)** Quantification of duration of GFP-hLC3A recruitment to entotic vacuoles during inner cell death in cells treated ± SaliP (2.5 µM). Data represent mean ± SD from five examples. **(i)** Widefield GFP-hLC3 and DIC time-lapse images of entotic cell-in-cell structures in MCF10A cells treated ± SaliP (2.5 µM). Asterisks denote inner cells and arrows denote GFP-hLC3A recruitment to entotic vacuoles. Scale bar, 10 µm; time, h:min. Source data are available for this figure: SourceData F5.

and influenza A virus infection (Ulferts et al., 2021). Strikingly, however, activation of canonical autophagy via mTOR inhibition (PP242) did not induce ATG16L1–V-ATPase interaction (Fig. 7, d and e), suggesting again that this mechanism is associated specifically with CASM. Finally, we found that treatment of cells with SaliP also promoted V-ATPase–ATG16L1 binding, dependent on K490A, indicating that enhanced V0–V1 association is sufficient to drive this interaction (Figs. 7 g and S4 d).

Together, these data reveal that V-ATPase recruits ATG16L1 during diverse non-canonical autophagy processes to drive CASM, although we do not yet know if this is through direct or indirect interaction. We speculate that V0–V1 association permits this interaction, perhaps via a conformational change, and that it requires the K490 containing pocket on the top face of the ATG16L1 C terminal WD40 β barrel. We propose that during CASM, the V-ATPase complex may play an analogous role to WIPI2 in canonical autophagy (Dooley et al., 2014), recruiting

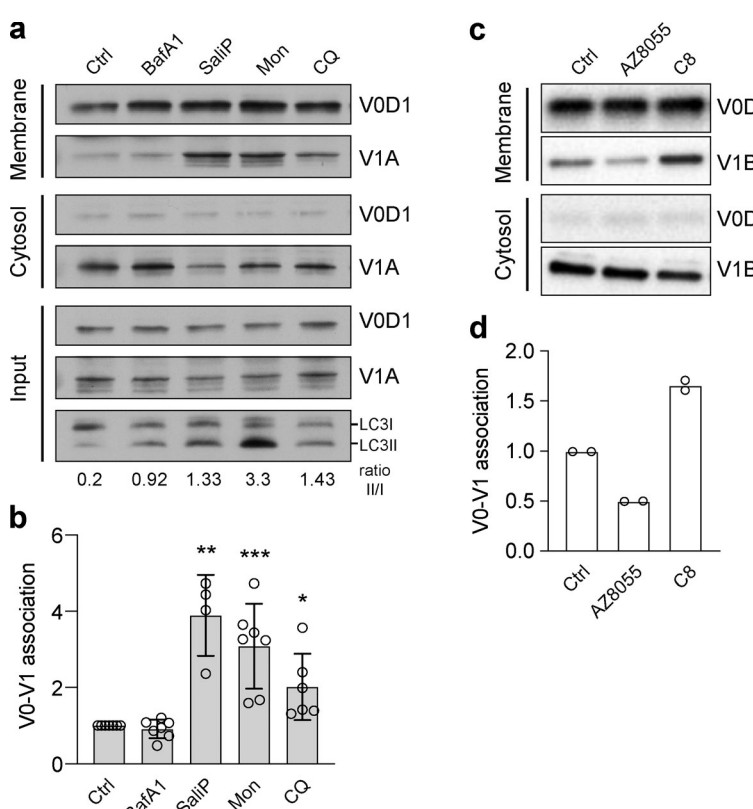

**Figure 6.** **Increased V0–V1 engagement is associated with activation of non-canonical autophagy. (a)** WT MCF10A cells were treated with BafA1 (100 nM), SaliP (2.5 μM), monensin (100 μM), or CQ (100 μM) for 1 h. Following fractionation, input, membrane, and cytosol fractions were probed for ATP6V1A, ATP6V0D1, and LC3 by Western blotting. LC3II/LC3I ratios are shown below. **(b)** Quantification of V0–V1 association from experiments as shown in a. Data represent mean ± SD from four to seven independent experiments. ***, P < 0.003; **, P < 0.01; *, P < 0.04, unpaired t test. **(c)** HeLa cells were stimulated with either mTOR inhibitor AZD8055 (1 μM) or TRPML1 agonist C8 (2 μM) for 90 min. Following fractionation, membrane and cytosol fractions were probed for ATP6V1B2, ATP6V0D1 by Western blotting. **(d)** Quantification of V0–V1 association from experiments as shown in c. Data represent mean ± SD from two independent experiments. Source data are available for this figure: SourceData F6.

ATG16L1 to the appropriate membrane, to specify the site of ATG8 conjugation.

### SopF blocks LAP and non-canonical autophagy

SopF is a *Salmonella* effector protein that blocks the interaction between V-ATPase and ATG16L1 during xenophagy by ribosylation of Gln124 in the ATP6V0C subunit (Xu et al., 2022; Xu et al., 2019). We reasoned that if V-ATPase drives LAP through recruitment of ATG16L1, then this would also be sensitive to SopF. Strikingly, transient overexpression of mCherry-SopF in RAW264.7 cells completely inhibited GFP-hLC3A lipidation during phagocytosis of zymosan (Fig. 8, a and b), without affecting ROS production in phagosomes, as determined by NBT test (Fig. 8 c). These data are consistent with a model in which ROS acts upstream of V-ATPase to modulate ATG16L1 recruitment. Interestingly, stable expression of SopF in MCF10A cells did not interfere with V0–V1 association, as promoted by monensin or SaliP (Fig. 8 d), suggesting it specifically blocks downstream ATG16L1 engagement. Notably, SopF also appears to inhibit non-canonical LC3 lipidation in the context of STING signaling, TRPML1 activation, and influenza A virus infection (Fischer et al., 2020; Goodwin et al., 2021; Ulferts et al., 2021). Taken together, our data, alongside these recent studies, support the conclusion that V-ATPase is a universal regulator of CASM, which functions to recruit ATG16L1 in a SopF-sensitive fashion.

### Non-canonical autophagy contributes to the *Salmonella* host response

Finally, we considered the mechanistic links between V-ATPase, ATG16L1, CASM, and SopF in the functional context of host-pathogen responses. *Salmonella* SopF provides a mechanism to evade host LC3 lipidation, and this has been elegantly studied in the context of xenophagy (Xu et al., 2019). Here, we investigated the hypothesis that SopF may also suppress CASM as a parallel host response. To explore a possible role for non-canonical autophagy during *Salmonella* infection, HCT116 cells, expressing WT or K490A ATG16L1, were infected with either WT or Δ*sopF* *Salmonella*. In agreement with Xu et al., we found GFP-rLC3B recruitment to *Salmonella* is increased using the Δ*sopF* strain in WT cells (Fig. 9, a–c). Importantly, we also found that in non-canonical autophagy deficient ATG16L1 K490A cells, GFP-rLC3B recruitment to either strain is dramatically reduced (Fig. 9, a–c). Considering the K490A mutation has no effect on canonical autophagy or autophagosome formation (Fletcher et al., 2018; Rai et al., 2019; Fig. 6 e), these data strongly suggest that *Salmonella* infection also induces a non-canonical autophagy response, which is disrupted by SopF targeting of the V-ATPase–ATG16L1 axis. Together, these findings indicate that *Salmonella* infection triggers CASM as a defensive host cell response, and that SopF has evolved as a mechanism of evasion.

## Discussion

Non-canonical autophagy, or CASM, has emerged as a pathway with vital functions in a wide range of biological processes, including immunity, vision, and cancer. A number of molecular players have been implicated in the regulation of this pathway, including the core ATG8 conjugation machinery, NADPH oxidase, Rubicon, and V-ATPase. However, exactly how these components work together and whether they are context-

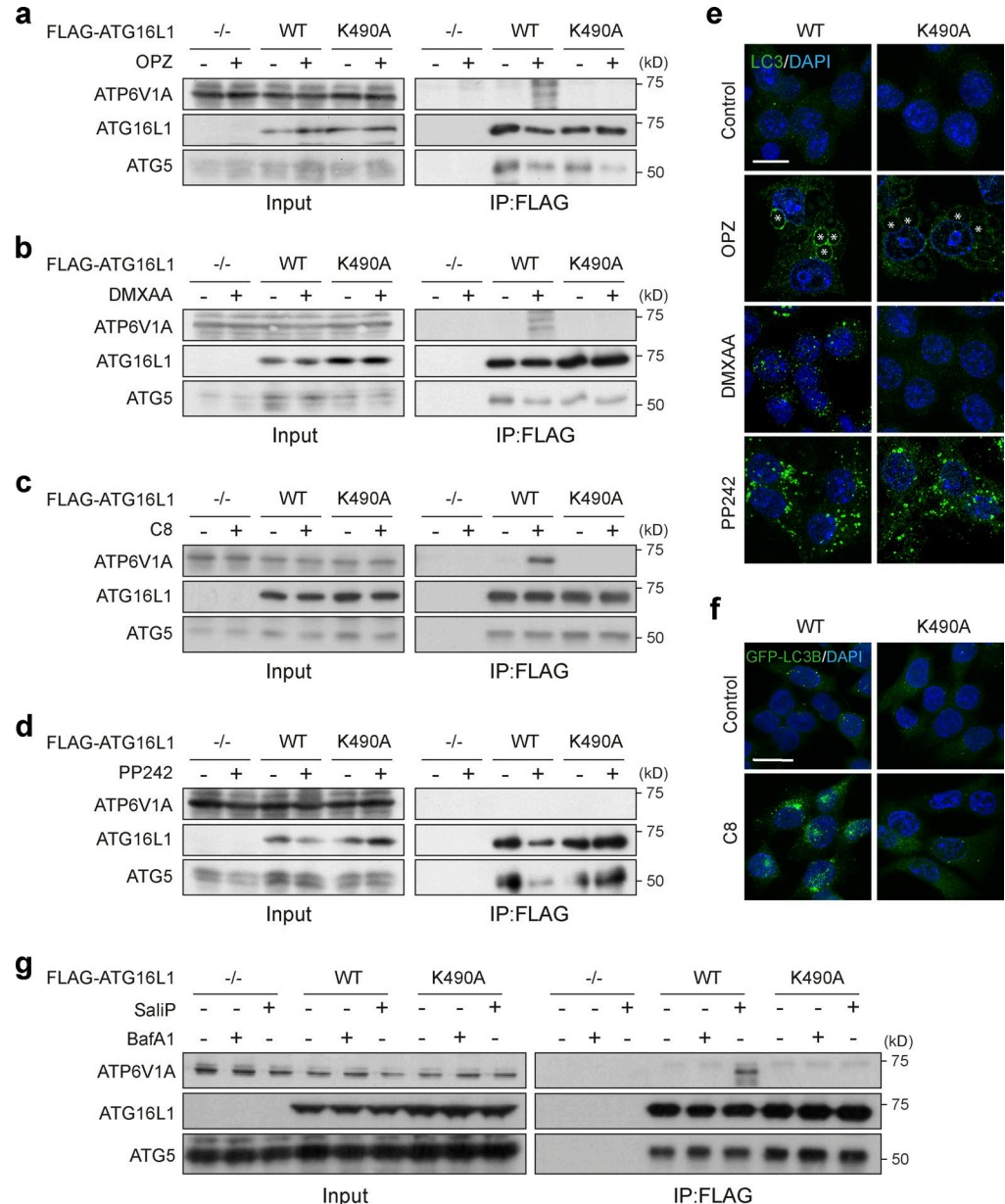

Figure 7. **ATG16L1 interacts with V-ATPase during non-canonical autophagy. (a and b)** *ATG16L1⁻/⁻* RAW264.7 cells and those re-expressing Flag-tagged WT and K490A ATG16L1, were treated with OPZ for 25 min (a) or STING agonist DMXAA (50 μg/ml) for 1 h (b). Input lysates and Flag immunoprecipitations were probed for ATP6V1A, ATG16L1, and ATG5 by Western blotting. **(c)** *ATG16L1⁻/⁻* HCT116 cells and those re-expressing FLAG-tagged WT and K490A ATG16L1 were treated with TRPML1 agonist C8 (2 μM) for 30 min. Input lysates and FLAG immunoprecipitations were probed by Western blotting as above. **(d)** *ATG16L1⁻/⁻* RAW264.7 cells and those re-expressing FLAG-tagged WT and K490A ATG16L1 were treated with mTOR inhibitor PP242 (1 μM) for 1 h. Input lysates and Flag immunoprecipitations were probed by Western blotting as above. **(e)** Confocal images of RAW264.7 cells expressing WT or K490A ATG16L1 treated with OPZ (25 min), DMXAA (1 h), or PP242 (1 h) and stained for LC3. Asterisks denote phagosomes. Scale bar, 10 μm. **(f)** Confocal images of GFP-rLC3B HCT116 cells expressing WT or K490A ATG16L1 treated with C8 (30 min). Scale bar, 15 μm. **(g)** *ATG16L1⁻/⁻* HCT116 cells and those re-expressing FLAG-tagged WT and K490A ATG16L1, were treated with BafA1 (100 nM) or SaliP (2.5 μM) for 1 h. Input lysates and FLAG immunoprecipitations were probed by Western blotting as above. Source data are available for this figure: SourceData F7.

specific or common to all forms of CASM has remained unclear. In this study, we focused on the involvement of the V-ATPase complex, its interplay with ATG16L1 during non-canonical autophagy processes, and its specific connection to NADPH oxidase/ROS during LAP.

V-ATPase was first implicated in non-canonical autophagy through pharmacological studies that demonstrated opposing effects of BafA1 and CQ on CASM during LAP and entosis (Florey et al., 2015). BafA1 and CQ are both lysosomal inhibitors, but with different mechanisms of action. BafA1 functions as a V-ATPase inhibitor and blocks CASM, while CQ, which acts instead to quench protons in the lysosome, induces CASM in a BafA1 sensitive fashion. These findings indicated that CASM responds to the lumenal ionic balance in lysosomes and that

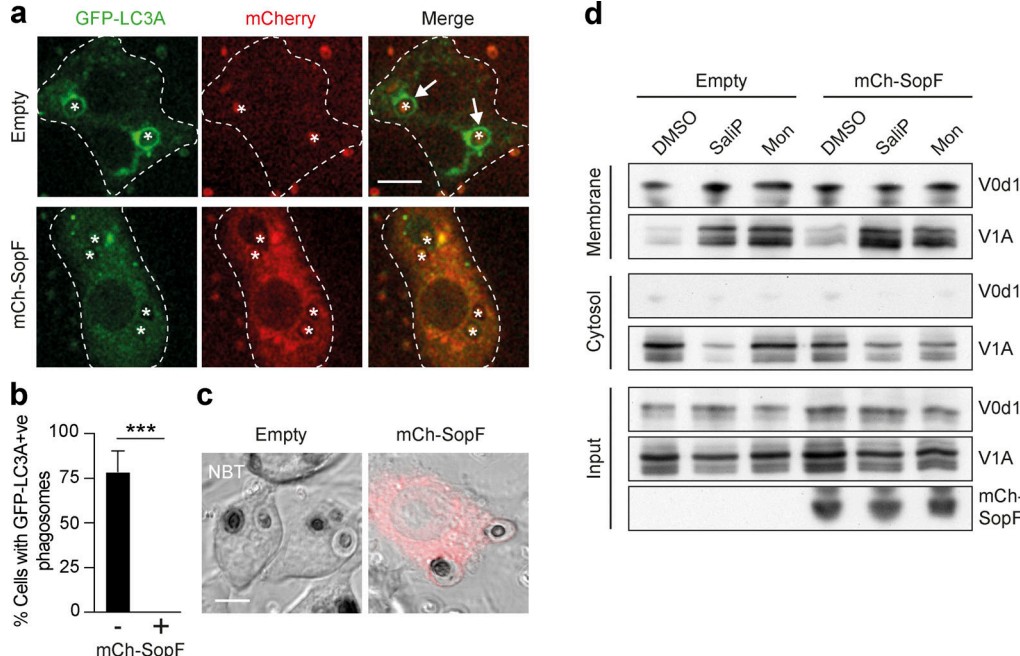

Figure 8. **The *Salmonella* effector protein SopF blocks LAP and non-canonical autophagy. (a)** Confocal images of GFP-hLC3A expressing RAW264.7 cells transfected with mCherry-SopF or empty vector and stimulated with OPZ for 25 min. Asterisks denote phagosomes and arrows mark GFP-LC3A positive phagosomes and dashed line marks outline of the cell. Scale bar, 5 µm. **(b)** Quantification of the percentage of phagocytosing mCherry-SopF or empty vector expressing cells that contain GFP-hLC3A positive phagosomes following OPZ stimulation. Data represent the mean ± SEM from three independent experiments. ***, P < 0.0003, unpaired *t* test. **(c)** Representative confocal DIC images of NBT/formazan deposits in phagosomes from empty vector and mCherry-SopF expressing RAW264.7 cells. Scale bar, 5 µm. **(d)** WT MCF10A cells expressing mCherry-SopF or empty vector were treated with SaliP (2.5 µM) or monensin (100 µM) for 1 h. Following fractionation, input, membrane, and cytosol fractions were probed for ATP6V1A, ATP6V0d1, and mCherry by Western blotting. Data representative of two repeats. Source data are available for this figure: SourceData F8.

V-ATPase is essential for the lipidation of ATG8s to single membranes during non-canonical autophagy.

This notion was somewhat unexpected, as V-ATPase is typically thought of as a downstream player during autophagy-related processes, driving terminal lysosomal degradation. However, an upstream, regulatory role for V-ATPase was further reinforced by the finding that, like CQ, a wide range of other ionophores and lysosmotropic agents, including monensin, nigericin, carbonyl cyanide 3-chlorophenylhydrazone, and clinically used drugs, such as lidocaine and amiodarone, similarly induce CASM in a BafA1 sensitive manner (Jacquin et al., 2017). Furthermore, a comparable pharmacological profile has also been observed during non-canonical autophagy following STING activation through a process thus termed VAIL (Fischer et al., 2020), and following TRPML1 activation (Goodwin et al., 2021).

Despite the growing evidence that V-ATPase is required to induce non-canonical autophagy, exactly how this complex might sense and respond to different stimuli and activate CASM remain unknown. In this study, we have defined a key regulatory mechanism (Fig. 10). Specifically, we find that neutralization of endolysosomal compartments leads to an increased association of the V-ATPase V0–V1 subunits, which in turn facilitates ATG16L1 recruitment through a K490-dependent interaction with the ATG16L1 C terminal WD domain. This mechanism operates in all tested examples of non-canonical autophagy, suggesting this represents a universal regulatory mechanism for the pathway.

Activation of non-canonical autophagy may thus represent a cellular response to stressed or perturbed endolysosomal compartments that transduce this signal via increased V0–V1 association. It seems likely that these stresses, including lumenal pH change or ionic imbalance, may involve different mechanisms depending on the specific stimulus. For example, pharmacological ionophores and lysosomotropic drugs directly neutralize endolysosomal compartments, which then increase V0–V1 association in an attempt to re-acidify; the ability of V-ATPase to sense and respond to altered lumenal pH in this way has been previously proposed (Marshansky, 2007). In a similar way, the influenza A virus M2 protein, which acts as a H⁺ proton ionophore, neutralizes intracellular vesicles to activate CASM (Fletcher et al., 2018; Ulferts et al., 2021).

During the activation of LAP, our data indicate that phagosome pH is neutralized by the high levels of ROS generated by NADPH oxidase. The ROS consume protons, thereby raising pH and driving increased phagosomal V1 levels in response. By linking together NADPH oxidase, ROS, and V-ATPase, these findings reveal direct connections between some of the key players in LAP, thus building toward a unifying mechanism. We note that phagosomal ROS have also been linked to the inhibition of ATG4 deconjugation activity during LAP (Ligeon et al., 2021), which may perhaps be related more to prolonging ATG8 lipidation at phagosomes than to its induction.

Further work will be required to assess the triggers for enhanced V0–V1 association in other contexts. In the case of

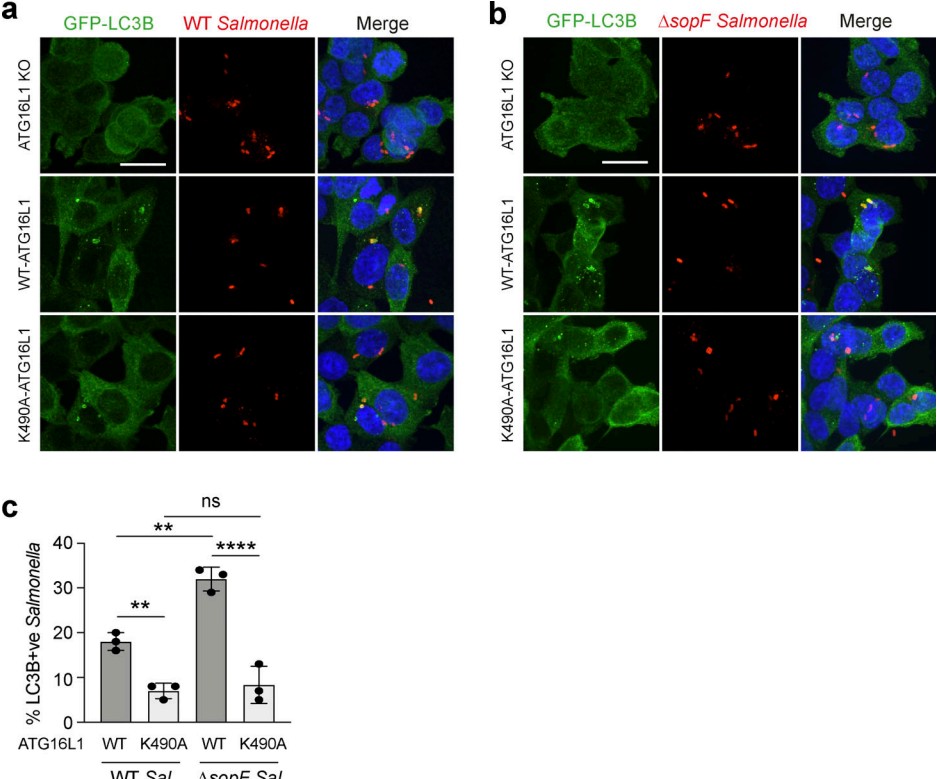

**Figure 9. Non-canonical autophagy contributes to the *Salmonella* host response. (a and b)** *ATG16L1⁻/⁻* HCT116 cells and those re-expressing FLAG-tagged WT and K490A ATG16L1, were infected with either WT (a) or *ΔsopF Salmonella* (b) for 1 h and imaged by confocal microscopy. Scale bar, 10 μm. **(c)** Quantification of GFP-rLC3B positive *Salmonella* in cells treated as in a and b. At least 100 bacteria were counted for each condition. Data represent mean ± SD from three independent experiments. **, P < 0.006; ****, P < 0.0001, one-way ANOVA followed by Tukey multiple comparison test.

entosis, which involves degradation of an entire internalized cell, it is tempting to speculate that the level of V-ATPase activity required to acidify such a large compartment may necessitate enhanced V0–V1 association. During VAIL, the trafficking of STING to endolysosomes or TBK1 phosphorylation may be involved. However, overall, it appears likely that while multiple

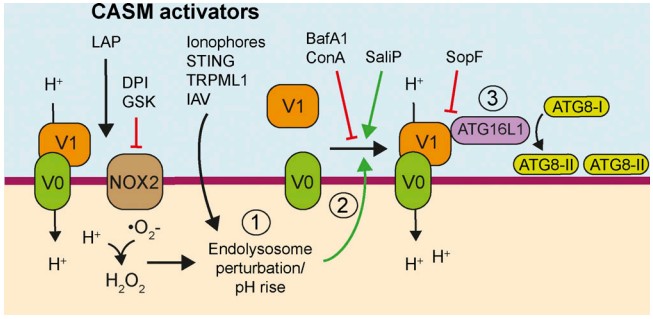

**Figure 10. Model of CASM activation.** (1) Stimuli that induce perturbations in endolysosomal ion and pH balance drive (2) V0–V1 engagement that (3) promotes the recruitment of ATG16L1 through its WD40 CTD and results in CASM. SopF modifies V-ATPase to inhibit ATG16L1 interaction while BafA1 and ConA interfere with V-ATPase activity, both resulting in inhibition of CASM. During LAP, NOX2-dependent ROS production consumes phagosomal H⁺ protons in a DPI and GSK sensitive manner, which alters phagosome pH and drives the SopF-sensitive interaction between V-ATPase and ATG16L1, which then directs ATG8 lipidation to phagosomes.

routes exist to trigger CASM at endolysosomal compartments, these all converge at V-ATPase, and specifically upon enhanced V0–V1 association, as a common step in non-canonical autophagy.

Importantly, an interaction between V-ATPase and the WD40 CTD of ATG16L1 was recently identified during *Salmonella* xenophagy (Xu et al., 2019). In this study, we report that this interaction is also induced across a range of non-canonical autophagy processes, including LAP, activation of TRPML1 or STING, and drug-induced CASM (monensin), but importantly, not during canonical autophagy. Currently, we do not know whether the interaction is direct or, if so, which part of the V-ATPase complex ATG16L1 binds to. We show that ATG16L1 interaction occurs upon enhanced V0–V1 association in a pump-independent manner, and indeed that increased V0–V1 engagement, triggered by SaliP, is sufficient to recruit ATG16L1. Furthermore, we map the interaction to a specific residue in the ATG16L1 WD40 CTD (K490). Together, these data indicate that the interaction between V-ATPase and ATG16L1 is universally required for non-canonical autophagy. We speculate that V-ATPase may perform a similar function to WIPI2 in canonical autophagy (Dooley et al., 2014), specifying the site of ATG16L1 recruitment to direct ATG8/LC3 lipidation during CASM. We note that additional processes such as regulation of LC3-dependent extracellular vesicle loading and secretion (Leidal et al., 2020) or ruffled border formation in osteoclasts (DeSelm

et al., 2011), also harness unconventional autophagy-related signaling. Whether these processes share the precise genetic and morphological features associated with CASM and harness the same V-ATPase–ATG16L1 axis remains to be determined.

The *Salmonella* effector SopF ribosylates Gln124 of ATP6V0c in the V-ATPase, inhibiting its interaction with ATG16L1 during xenophagy (Xu et al., 2022; Xu et al., 2019) and also during STING activation (Fischer et al., 2020), influenza virus A infection, and a range of other CASM processes, as shown here. Further to this, our data show that while SopF blocks CASM, it does not block increased V0–V1 association. These findings suggest a specific sequence of events during CASM (Fig. 10) and the existence of a SopF-sensitive conformational change that occurs during increased V-ATPase activity and V0–V1 association to engage ATG16L1. Using cryo-EM, recent advances have been made in determining the complete mammalian structure of V-ATPase (Wang et al., 2020a; Wang et al., 2020b), where different holoenzyme states have been identified. In the future, it may be possible to resolve specific structural features in SopF-modified versus unmodified complexes that are required to support ATG16L1 WD40 CTD binding.

V-ATPase activity is well known to be regulated along the endocytic pathway, with increased assembly and lower pH as it progresses toward lysosomes (Lafourcade et al., 2008). However, the molecular mechanisms regulating V0–V1 association in mammalian cells remain poorly understood. Glucose and amino acid availability can modulate the assembly and association of the V-ATPase complex (McGuire and Forgac, 2018; Stransky and Forgac, 2015) in a PI3-kinase dependent manner, and lipid composition is proposed to play a regulatory role, with PI(3,5)P2 stabilizing the V0–V1 association (Li et al., 2014). Changes in V0–V1 levels have also been reported in different differentiation states of murine dendritic cells, with more mature cells displaying increased V0–V1 association (Liberman et al., 2014). Our data now reveal an increased V1A recruitment to phagosomes in BMDCs compared with BMDM. Thus, it seems likely that differences in non-canonical autophagy activation may be observed between different cell types and species, dependent on basal differences in their regulation of V-ATPase.

While non-canonical autophagy regulates a wide range of fundamental biological processes (Heckmann et al., 2017), the underlying molecular and cellular functions of CASM are still not fully understood. During LAP, ATG8 recruitment has been proposed to modulate phagosome maturation and content degradation. However, while some studies suggest a role for ATG8s in promoting phagosome maturation and lysosome fusion (Gluschko et al., 2018; Henault et al., 2012; Martinez et al., 2015), others suggest instead that ATG8 recruitment delays maturation to stabilize phagosomes (Romao et al., 2013). In another study, no gross defect in phagosome maturation was detected upon inhibition of ATG8 lipidation (Cemma et al., 2016), and, BafA1, which inhibits LAP, does not impair the recruitment of LAMP1 to phagosomes (Kissing et al., 2015). Considering our new data, which show that ATG8/LC3 is recruited to phagosomes after the acquisition of late endosome/lysosome markers, it seems possible that LAP may act to fine tune, rather than initiate, phagosome maturation. For instance, perhaps ATG8/LC3s recruit

interacting proteins to modulate the extent of lysosome fusion and/or signaling from phagosomes (Manil-Ségalen et al., 2014; McEwan et al., 2015). Through γ-aminobutyric acid receptor-associated protein conjugation to endosomes, lysosome biogenesis is also enhanced via transcription factor EB transcription factor activity (Goodwin et al., 2021). Further work will be required to resolve these debates in the literature and to elucidate possible cell type or context-specific differences.

Non-canonical autophagy and the ATG16L1–V-ATPase axis facilitate a key innate immune response to pathogen infection. As such, pathogens have evolved evasion strategies, which further underscore the functional importance of CASM. The intracellular pathogen, *Legionella pneumophila*, expresses an effector protein, RavZ, which deconjugates ATG8 from both PE and PS (Durgan et al., 2021), and blocks LAP (Martinez et al., 2015). Cell wall melanin from the fungal pathogen, *Aspergillus fumigatus*, inhibits LAP by interfering with NADPH oxidase and ROS production (Akoumianaki et al., 2016; Kyrmizi et al., 2018). As noted above, the *Salmonella* effector protein, SopF, directly targets the ATG16L1–V-ATPase interaction to permit bacterial growth (Xu et al., 2019), and to date, this effect has been attributed solely to xenophagy inhibition. Here, using genetic models to specifically inhibit non-canonical autophagy, we build on these findings to show that SopF also inhibits CASM, and suggest that the ATG8 response to *Salmonella* is, in part, due to non-canonical autophagy. These data are consistent with other studies implicating LAP in *Salmonella* infection (Masud et al., 2019). In the future, it will be important to explore the impacts of other pathogen effectors on CASM.

In conclusion, we propose a key molecular mechanism unifying the various forms of non-canonical autophagy, whereby enhanced V0–V1 association recruits ATG16L1 to V-ATPase positive, endolysosomal compartments to drive CASM. This mechanism explains the unusual pharmacological profile of CASM processes and provides new molecular insight into how NADPH oxidase and ROS activate LAP. In the future, it will be of interest to test this model against further stimuli of non-canonical autophagy. These findings also build on the recent elucidation of V-ATPase–ATG16L1 binding during *Salmonella*-induced xenophagy, indicating that this interaction also occurs during non-canonical autophagy, and indeed that V-ATPase specifies the site of single-membrane ATG8 lipidation during CASM. Finally, our data implicate CASM as a parallel pathogen response during *Salmonella* infection, blocked by the effector protein SopF.

## Materials and methods
### Antibodies
Antibodies used were rabbit polyclonal anti-ATG16L1 (8090; Cell Signaling Technology, (Western blotting [WB] 1:1,000), rabbit polyclonal anti-ATG5 (2630; Cell Signaling Technology, WB 1:1,000), rabbit polyclonal anti-LC3A/B (4108; Cell Signaling Technology, WB 1:1,000, immunofluorescence [IF] 1:100), rabbit monoclonal anti-ATP6V1A (ab199326; Abcam, WB 1:2,000, IF 1:250), mouse anti-ATP6V1B1/2 (sc55544; SCBT, WB 1:1,000), mouse monoclonal anti-ATP6V0d1 (ab56441; Abcam, WB 1:

1,000), mouse monoclonal anti-LAMP1 (555798; BD Biosciences, IF 1:100), mouse monoclonal anti-mCherry (ab125096; Abcam, WB 1:1,000), mouse monoclonal anti-GAPDH (ab8245; Abcam, WB 1:1,000), Alexa Fluor 488 polyclonal goat anti-rabbit IgG (A-11034; Thermo Fisher Scientific, IF 1:500), Alexa Fluor 568 polyclonal goat anti-mouse IgG (A-11004; Thermo Fisher Scientific, IF 1:500), Alexa Fluor 568 polyclonal goat anti-rabbit IgG (A-11011; Thermo Fisher Scientific, IF 1:500), HRP-conjugated anti-rabbit IgG (7074; Cell Signaling Technology, WB 1:2,000), and HRP-conjugated anti-mouse IgG (7076; Cell Signaling Technology, WB 1:2,000).

## Reagents

Reagents and chemicals used were BafA1 (1334; Tocris, 100 nM), PP242 (4257; Tocris, 1 μM), AZD8055 (S1555; Selleckchem, 1 μM), Monensin (M5273; Sigma-Aldrich, 100 μM), DPI (D2926; Sigma-Aldrich, 5 μM), human serum (H2918; Sigma-Aldrich), DMXAA (D5817; Sigma-Aldrich, 50 mg/ml), Zymosan (Z4250; Sigma-Aldrich), NBT (N6876; Sigma-Aldrich), murine IFNγ (315-05; PeproTech), DAPI (D9542; Sigma-Aldrich), IN-1 (17392; Cayman Chemical, 1 μM), LTR DND-99 (L7528; Thermo Fisher Scientific), LysoTracker Deep Red (L12492; Thermo Fisher Scientific), E64d (E8640; Sigma-Aldrich, 10 μg/ml) anti-FLAG M2 magnetic beads (M8823; Sigma-Aldrich). SaliP (2.5 μM) and TRPML1 agonist C8 (2 μM) were provided by Casma Therapeutics.

## Plasmid constructs

pmCherry-SopF was generated previously (Lau et al., 2019) and kindly provided by Dr. Leigh Knodler, Washington State University, Pullman, WA. mCherry-SopF was cloned into pQCXIP using AgeI/BamHI restriction sites. mRFP-Rab7 was a gift from Ari Helenius (plasmid #14436; Addgene). pBabe-Puro-RFP-LAMP1 was kindly provided by Dr. Michael Overholtzer, Memorial Sloan-Kettering Cancer Center, New York, NY.

## Cell culture

WT or $ATG13^{-/-}$ MCF10A cells (human breast epithelial), expressing GFP-LC3A (human), were prepared as described previously (Jacquin et al., 2017) and cultured in DMEM/F12 (11320074; Gibco) containing 5% horse serum (16050-122; Thermo Fisher Scientific), EGF (20 ng/ml; AF-100-15; Pepro-Tech), hydrocortisone (0.5 mg/ml; H0888; Sigma-Aldrich), cholera toxin (100 ng/ml; C8052; Sigma-Aldrich), insulin (10 μg/ml; I9278; Sigma-Aldrich), and penicillin/streptomycin (100 U/ml; 15140-122; Gibco) at 37°C, 5% $CO_2$.

HCT116 cells (human colorectal epithelial) were maintained using DMEM (41966-029; Gibco) supplemented with 10% FBS (F9665; Sigma-Aldrich) and penicillin/streptomycin (P/S; 100 U/ml, 100 μg/ml; 15140-122; Gibco) at 37°C, 5% $CO_2$. A panel of HCT116 lines expressing GFP-LC3B (rat) and different ATG16L1 constructs were derived from $ATG16L1^{-/-}$ cells, reconstituted with pBabe-Puro FLAG-S-ATG16L1 (WT or K490A), as described previously (Fletcher et al., 2018).

RAW264.7 cells were obtained from American Type Culture Collection and maintained in DMEM (41966-029; Gibco) supplemented with 10% FBS (F9665; Sigma-Aldrich) and penicillin/

streptomycin (100 U/ml, 100 μg/ml; 15140-122; Gibco) at 37°C, 5% $CO_2$. $ATG16L1^{-/-}$ were generated and reconstituted with pBabe-Puro FLAG-S-ATG16L1 (WT or K490A) and GFP-LC3A (human), as described previously (Durgan et al., 2021; Lystad et al., 2019).

## Retrovirus generation and infection

For mCherry-SopF virus infection, MCF10A cells were seeded in a 6-well plate at $5 \times 10^4$ per well. The next day 1 ml viral supernatant was added with 10 μg/ml polybrene for 24 h followed by a media change. Cells were then selected with puromycin (2 μg/ml).

## BMDC and BMDM isolation

Bone marrow extracted from the hind legs of one C57/BL6 mouse was resuspended in 10 ml complete RPMI 1640 media (R8578; Sigma-Aldrich; 10% FBS, 1% P/S, 50 μM 2-mercaptoethanol), then centrifuged at 1,500 rpm for 5 min. Bone marrow was then resuspended in 1 ml of RBC lysis buffer (168 mM $NH_4Cl$, 100 μM $KHCO_3$, 10 mM $Na_2EDTA$ [pH 8], in Milli-Q $H_2O$, final pH 7.3) and agitated for 2 min prior to resuspension in 10 ml complete RPMI 1640 media and centrifuged at 1,500 rpm for 5 min. For DC differentiation, bone marrow was resuspended in 25 ml of complete RPMI 1640 media supplemented with 20 ng/ml GM-CSF (#315-03; PeproTech), 10 ng/ml IL-4 (#AF-214-14; PeproTech), and 50 ng/ml Amphotericin B (#15290018; Gibco). For macrophage differentiation, bone marrow was resuspended in 25 ml of complete RPMI 1640 media supplemented with 20 ng/ml M-CSF (#AF-315-02; PeproTech) and 50 ng/ml Amphotericin B. The cells were incubated for 3 d at 37°C, 5% $CO_2$, in low-adherence 90-mm sterilin Petri dishes (101R20; Thermo Fisher Scientific). On the third day, media was removed and centrifuged at 1,500 rpm for 5 min. The non-adherent cells were then resuspended in fresh differentiation media and added back to the dishes containing the adherent cells for a further 3 d of differentiation. On the sixth day, non-adherent cells in the media were removed and adherent cells were harvested by gently scraping into complete RPMI 1640 media. The cells were then centrifuged at 1,500 rpm for 5 min and seeded in complete RPMI 1640 media without differentiation cytokines.

## Amaxa nucleofection

Transient transfection was performed by electroporation using a Nucleofector II instrument (Lonza) and Lonza nucleofection kit V (VCA-1003; Lonza) following the manufacturer's guidelines. Briefly, $2 \times 10^6$ RAW264.7 were electroporated with 2 μg mCherry-SopF or RFP-Rab7 using program D-032.

## Fluorescent live cell microscopy and analysis

Prior to imaging, RAW264.7 cells were plated overnight on 35 mm glass-bottomed dishes (MatTek) in the presence of 200 U/ml IFNγ. Opzonised zymosan (OPZ) particles were generated by incubating Zymosan A from Saccharomyces (Z4250; Sigma-Aldrich) in human serum (P2918; Sigma-Aldrich) for 30 min at 37°C. The zymosan was then centrifuged at 5,000 rpm for 5 min and then resuspended in PBS at 10 mg/ml. The solution was passed through a 25-gauge needle using a 1-ml syringe several

times to break up aggregates. Cells were mounted on a spinning-disk confocal microscope, comprising a Nikon Ti-E stand, Nikon 60 × 1.45 NA oil immersion lens, Yokogawa CSU-X scanhead, Andor iXon 897 EM-CCD camera, and Andor laser combiner. All imaging with live cells was performed within incubation chambers at 37°C and 5% $CO_2$. For RFP-Rab7, LAMP1-RFP, LTR and GFP-LC3 imaging, z stacks were acquired every 30 s following addition of OPZ. Image acquisition and analysis were performed with Andor iQ3 (Andor Technology) and ImageJ.

For analysis of GFP-LC3 recruitment to phagosomes, IFNγ-treated RAW264.7 cells were pre-treated with inhibitors as indicated, followed by stimulation with OPZ for 25 min at 37°C. Z-stacks from multiple fields of view were acquired by spinning-disk confocal microscopy as described above. ImageJ software was used to measure the mean GFP intensity in a region of interest at individual phagosome membranes and divided by the intensity of a region of interest within the cytosol. Or, maximum GFP intensity from line profiles made over multiple individual phagosomes from three independent experiments, were measured using ImageJ software.

For lysotracker imaging, IFNγ-treated RAW264.7 cells were incubated with LTR (50 nM) for 15 min at 37°C. Z stacks were acquired over time or after 25 min stimulation with OPZ by spinning-disk microscopy as described above. Mean fluorescent intensity of individual phagosomes was measured using ImageJ software. For primary cells, Lysotracker Deep Red (50 nM) was used and images were acquired using a Zeiss LSM 780 confocal microscope (Carl Zeiss Ltd) using Zen software (Carl Zeiss Ltd).

For entosis, MCF10A cells were plated on glass-bottomed 6-well plates (MatTek). The next day, cells were maintained in an incubation chamber at 37°C and 5% $CO_2$, and cell-in-cell structures were imaged by widefield time-lapse microscopy. Fluorescent and differential interference contrast (DIC) images were acquired every 8 min using a Flash 4.0 v2 sCMOS camera (Hamamatsu), coupled to a Nikon Ti-E inverted microscope, using a 20 × 0.45 NA objective. Image acquisition and analysis were performed with Elements software (Nikon).

### Fixed immunofluorescence confocal microscopy and analysis
Cells were seeded in 12-well plates containing coverslips and incubated at 37°C, 5% $CO_2$ for 24 h. Following treatments, cells were washed twice with ice-cold PBS and then incubated with 100% methanol at –20°C for 10 min. The cells were then washed twice with PBS and blocked with 0.5% BSA (A7906; Sigma-Aldrich) in PBS for 1 h at RT. The cells were incubated overnight at 4°C with the primary antibodies and then washed three times in cold PBS. Fluorescent secondary antibodies were used at a 1:500 dilution in PBS + 0.5% BSA and were incubated with the cells for 1 h at RT. The cells were washed three times in cold PBS prior to being incubated with DAPI for 10 min at RT and then mounted onto microscope slides with ProLong Gold anti-fade reagent (P36930; Invitrogen). Image acquisition was made using a Zeiss LSM 780 confocal microscope (Carl Zeiss Ltd), using Zen software (Carl Zeiss Ltd).

For primary cells, maximal ATP6V1A intensity was quantified at the phagosome membrane and divided by the mean intensity within the cell perimeter using ImageJ software. For each condition, 10 phagosomes within four images were quantified and repeated in three independent experiments. For RAW264.7 cells, maximum intensity from line profiles made over individual phagosomes was measured from three independent experiments using ImageJ software.

### Salmonella infection
Infections were performed with WT and ΔsopF Salmonella enterica serovar Typhimurium SL1344 which are kind gifts from Dr. Feng Shao. Late-log bacterial cultures were used for infection during experiments as outlined previously (D'Costa et al., 2015). Briefly, GFP-LC3 expressing HCT116 cells were seeded on 1-cm² glass coverslips in 24-well tissue culture plates 48 h before use at a density of 100,000 cells/coverslip. Bacteria were grown for 16 h at 37°C with shaking and then subcultured (1:33) in Lysogeny broth (LB) without antibiotics for 3 h. After subculture, bacteria were pelleted at 10,000g for 2 min, resuspended and diluted 1:100 in PBS, pH 7.2, with calcium and magnesium, and added to cells for 10 min at 37°C. The cells were then washed three times with PBS with calcium and magnesium. Selection for intracellular bacteria was performed 30 min after infection using 100 µg/ml gentamicin. At 1 h after infection, cells were fixed with 4% PFA in PBS at 37°C for 10 min.

### NBT/formazan assay
RAW264.7 cells plated on coverslips were incubated with 0.2 mg/ml NBT for 10 min at 37°C. Where indicated, cells were also preincubated with inhibitors prior to addition of OPZ for 25 min. Samples were then fixed by ice-cold methanol, and dark formazan deposits were detected by DIC imaging on a Zeiss LSM 780 confocal microscope (Carl Zeiss Ltd) using Zen software (Carl Zeiss Ltd).

### Luminol assay
RAW264.7 cells were seeded in 96-well white microwell plates (136101; Thermo Fisher Scientific) at 1 × 10⁵ cells/well and incubated at 37°C, 5% $CO_2$ for 24 h. Growth media was removed and cells were washed with PBS. Any pretreatments were incubated at 37°C, and a final concentration of 0.32 U/ml of HRP (P8375; Sigma-Aldrich) and a final concentration of 125 nM of luminol (A8511; Sigma-Aldrich) diluted in DPBS (Dulbecco's PBS with $Ca^{2+}$ and with $Mg^{2+}$, D8662; Sigma-Aldrich) supplemented with 45% glucose solution (G8769; Sigma-Aldrich) and 7.5% sodium bicarbonate solution (S8761; Sigma-Aldrich) were added to the cells and incubated for the final 3 min of the pretreatment incubation at 37°C. Stimuli were then added in triplicate and measurements were acquired immediately in the Micro-LumatPlus LB 96V (Berthold technologies) at 37°C.

### Membrane fractionation
MCF10A cells were seeded on a 15-cm dish and cultured for 48 h. Cells were stimulated in suspension as indicated for 1 h. Input, cytosol, and membrane fractions were isolated using the Mem-Per Plus Membrane Protein Extraction Kit (89842; Thermo Fisher Scientific) following product guidelines. Protein concentration was measured by BCA assay and equal amounts were loaded onto polyacrylamide gels for SDS–PAGE analysis.

## Immunoprecipitation

One confluent 15-cm plate of HCT116 and RAW 264.7 cells was used per condition. Following stimulation, cells were washed twice with ice-cold PBS then lysed in 700 µl of lysis buffer (50 mM Tris-HCl [pH 7.5–7.6], 150 mM NaCl, 2 mM EDTA, 0.8% C12E9, and protease/phosphatase inhibitors) per plate, on ice, and incubated for 20 min at 4°C. Lysates were then centrifuged at 13,500 rpm for 10 min at 4°C and then rotated with 10 µl of pre-equilibrated protein A beads per sample (Pierce Protein A Magnetic Beads, 88845; Thermo Fisher Scientific, RA228548) for 1 h at 4°C. Following separation by magnet, lysates were then rotated with 25 µl of pre-equilibrated anti-FLAG M2 Magnetic Beads per sample (M8823; Millipore) for 2 h at 4°C. The lysates were washed five times in adjusted lysis buffer (50 mM Tris-HCl [pH 7.5–7.6], 150 mM NaCl, 2 mM EDTA, 0.1% C12E9, and 1% Triton X-100) by rotating for 10 min at 4°C. Samples were eluted by boiling at 95°C in 25 µl 2× SDS-buffer for 10 min.

## Western blotting

Cells were scraped into ice-cold radioimmunoprecipitation assay buffer (150 mM NaCl, 50 mM Tris–HCl, pH 7.4, 1 mM EDTA, 1% Triton X-100 (T8787; Sigma-Aldrich), 0.1% SDS (L3771; Sigma-Aldrich), 0.5% sodium deoxycholate (D6750; Sigma-Aldrich), and lysed on ice for 10 min. Lysates were centrifuged for 10 min at 10,000$g$ at 4°C. Supernatants were then separated on 10% or 15% polyacrylamide SDS–PAGE gels and transferred to polyvinylidene difluoride membranes. Membranes were blocked in TBS-Tween supplemented with 5% BSA for 1 h at RT and incubated overnight at 4°C with primary antibodies diluted in blocking buffer. They were then incubated with an HRP-conjugated secondary antibody, and proteins were detected using ECL (RPN2209; GE Healthcare Life Sciences) and developed using an MI-5 X-Ray film processor (Medical Index). Densitometry analysis of bands was performed using ImageJ software.

## Statistical analysis

Statistical analysis was performed using GraphPad Prism software. One-way ANOVA followed by Tukey multiple comparison test or unpaired $t$ test were used as indicated in figure legends.

## Online supplemental material

Fig. S1 shows GFP-LC3A and RFP-LAMP1 dynamics during entosis. Fig. S2 shows LysoTracker intensity of phagosomes treated with BafA1 or DPI. Fig. S3 shows effect of E64d on DPI inhibition of LAP. Fig. S4 provides additional data on ATG16L1-V-ATPase interaction. Video 1 shows GFP-LC3A and Rab7-RFP dynamics during phagocytosis. Video 2 shows GFP-LC3A and LAMP1-RFP dynamics during phagocytosis. Video 3 shows effect of SaliP on GFP-LC3 dynamics during entosis.

## Acknowledgments

We thank Nick Ktistakis, Simon Cook, and Phill Hawkins for helpful discussions and critical review of the manuscript, and the BI Imaging facility for all their support.

This work was supported by grants from the Biotechnology and Biological Sciences Research Council, BB/P013384/1 (BBS/E/ B/000C0432 and BBS/E/B/000C0434), BB/R019258/1; and Cancer Research UK Career Development award C47718/A16337. J.H. Brumell holds the Pitblado Chair in Cell Biology. Work in the lab of J.H. Brumell was supported by an operating grant from the Canadian Institutes of Health Research (FDN#154329).

O. Florey is an ad hoc paid consultant for Casma Therapeutics. J.M. Goodwin is an employee and shareholder of Casma Therapeutics. The authors declare no further competing financial interests.

Author contributions: K.M. Hooper and E. Jacquin designed and carried out experiments. T. Li and J.H. Brumell carried out and analyzed *Salmonella* experiments. J.M. Goodwin provided reagents and insights and performed preliminary experiments. J. Durgan designed experiments and wrote the manuscript. O. Florey designed and carried out experiments and wrote the manuscript. All authors read and reviewed the manuscript.

Submitted: 25 May 2021

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

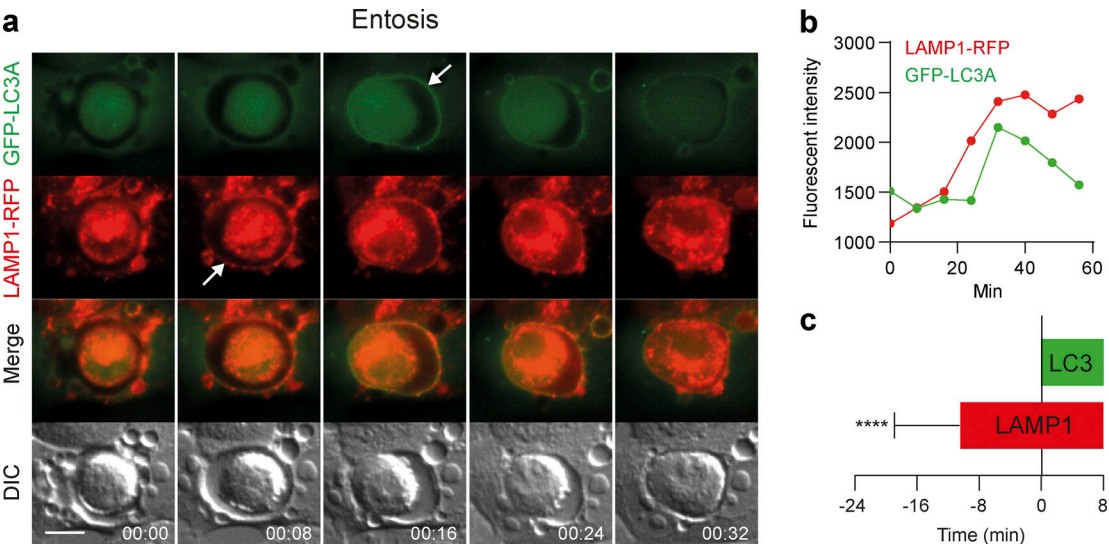

Figure S1. **LAMP1 recruits to entotic vacuoles prior to LC3. (a)** Representative time-lapse widefield microscopy images of entosis in MCF10A cells expressing GFP-hLC3A and LAMP1-RFP. Arrows mark acquisition of fluorescent markers. Scale bar, 15 µm; time, min:s. **(b)** Analysis of fluorescent marker intensity at the entotic vacuole membrane over time. **(c)** Quantification of LAMP1-RFP acquisition from 23 entotic vacuoles in relation to GFP-hLC3A (time 0). Data represent mean ± SD. ****, P < 0.0001, unpaired *t*-test.

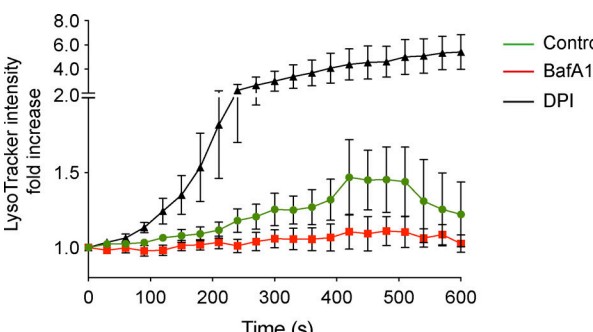

Figure S2. **Confocal microscopy measurements of phagosome LysoTracker intensity over time in RAW264.7 cells pre-treated with either DPI (5 µM) or BafA1 (100 nM) prior to addition of OPZ.** Data represent mean values normalized to time 0 ± SEM of nine individual phagosomes across multiple independent experiments.

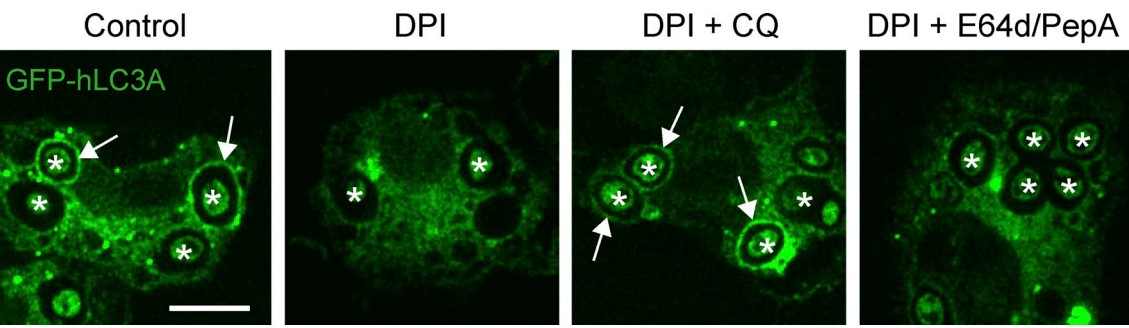

**Figure S3. Effect of E64d treatment on LAP.** Confocal images of GFP-hLC3A expressing RAW264.7 cells treated with DPI (5 mM), CQ (100 mM), or E64d + pepstatin (10 µg/ml) as indicated, followed by addition of OPZ for 25 min. Asterisks denote phagosomes and arrows mark GFP-LC3A positive phagosomes. Scale bar, 5 µm.

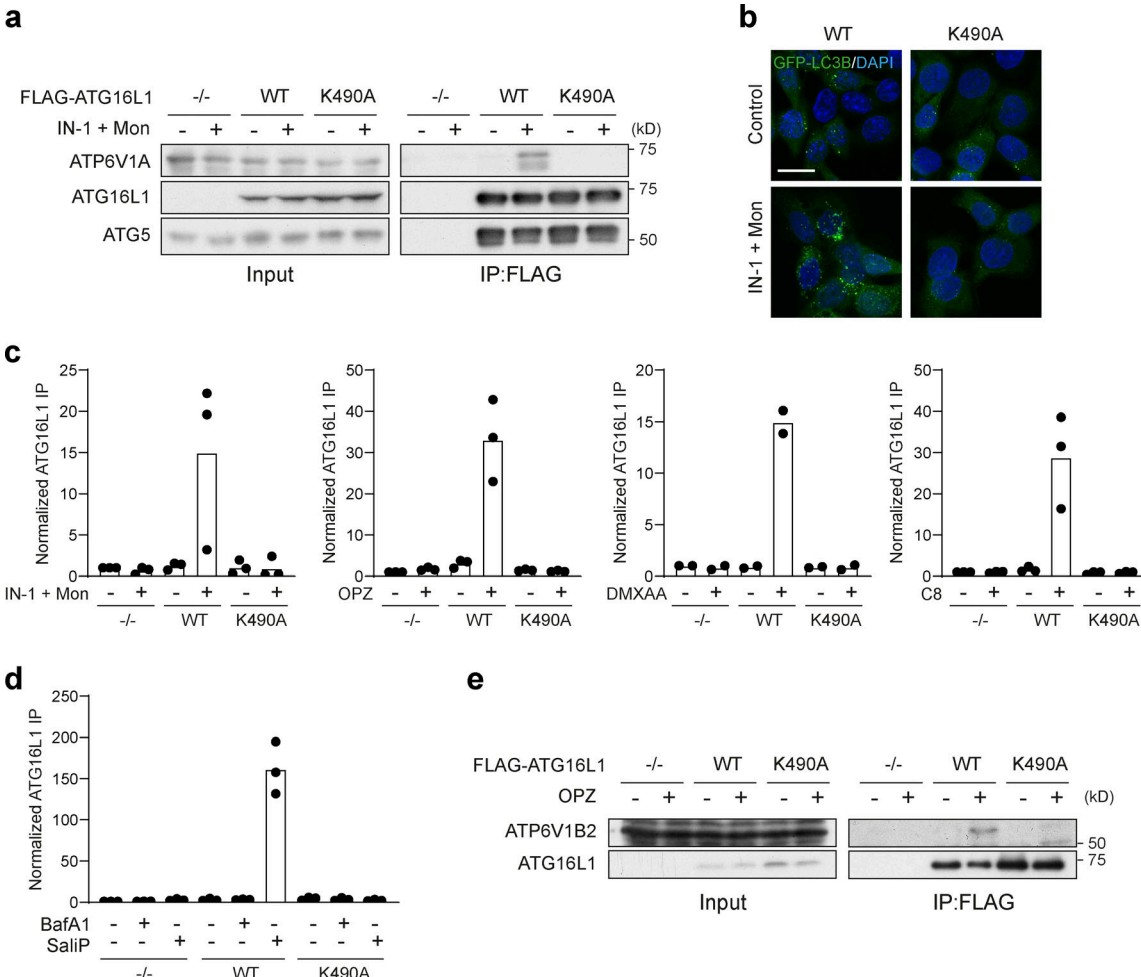

**Figure S4. Quantification of FLAG-ATG16L1 co-immunoprecipitation of V-ATPase subunits. (a)** *ATG16L1*−/− HCT116 cells and those re-expressing FLAG-tagged WT and K490A ATG16L1 were treated with monensin (100 µM) + IN-1 (1 µM) for 45 min. Input lysates and FLAG immunoprecipitations were probed for ATP6V1A, ATG16L1, and ATG5 by Western blotting. **(b)** Confocal images of GFP-rLC3B HCT116 cells expressing WT or K490A ATG16L1 treated as in a. Scale bar, 15 µm. **(c and d)** Quantification of ATP6V1A pulldown in Flag-ATG16L1 immunoprecipitations as shown in Fig. 7, a–d, and Fig. S4 a. Data represent means normalized to unstimulated controls from two to three independent experiments. **(e)** *ATG16L1*−/− RAW264.7 cells and those re-expressing Flag-tagged WT and K490A ATG16L1 were treated with OPZ for 25 min. Input lysates and Flag immunoprecipitations were probed for ATP6V1B2 and ATG16L1 by Western blotting. Source data are available for this figure: SourceData FS4.

Video 1. **Time-lapse spinning-disk confocal microscopy showing phagocytosis of OPZ in RAW264.7 cells expressing GFP-hLC3A and RFP-Rab7.** Sequential recruitment of RFP-Rab7 and hGFP-LC3A to phagosomes is observed. Image stacks acquired every 30 s. Movie plays 4 frames/s; time, min:s. Related to Fig. 1 a.

Video 2. **Time-lapse spinning-disk confocal microscopy of showing phagocytosis of OPZ in RAW264.7 cells expressing GFP-hLC3A and LAMP1-RFP.** Sequential recruitment of LAMP1-RFP and hGFP-LC3A to phagosomes is observed. Image stacks acquired every 30 s. Movie plays 3 frames/s; time, min:s. Related to Fig. 1 b.

Video 3. **Time-lapse widefield fluorescent microscopy of entotic cell-in-cell structures in MCF10A cells expressing GFP-hLC3A treated ± SaliP (SaliPhe; 2.5 μM).** Movie shows GFP-hLC3A recruitment to entotic vacuoles during inner cell death. GFP-hLC3A recruitment is prolonged in the presence of SaliP. GFP and DIC images acquired every 10 min for 20 h. Movie plays 8 frames/s; time, h:min. Related to Fig. 5 i.

