## [Peer Review File · The Journal of Cell Biology]

V-ATPase is a universal regulator of LC3-associated phagocytosis and non-canonical autophagy

Kirsty Hooper, Elise Jacquin, Taoyingnan Li, Jonathan Goodwin, John Brumell, Joanne Durgan, and Oliver Florey

Corresponding Author(s): Oliver Florey, Babraham Institute

Review Timeline:

Submission Date:	2021-05-25
Editorial Decision:	2021-07-13
Revision Received:	2022-02-04
Editorial Decision:	2022-03-07
Revision Received:	2022-04-12

Monitoring Editor: Tamotsu Yoshimori

Scientific Editor: Andrea Marat

Transaction Report:

DOI: <https://doi.org/10.1083/jcb.202105112>

July 13, 2021

Re: JCB manuscript #202105112

Dr. Oliver Florey
Babraham Institute
Signalling Programme The Babraham Institute
Cambridge CB22 3AT
United Kingdom

Dear Dr. Florey,

Thank you for submitting your manuscript entitled "V-ATPase is a universal regulator of LC3 associated phagocytosis and non-canonical autophagy". The manuscript was assessed by expert reviewers, whose comments are appended to this letter. We invite you to submit a revision if you can address the reviewers' key concerns, as outlined here.

You will see that the reviewers find that your study provides interesting insight into the process of CASM, and we agree that a suitably revised study is of high interest for the readership of JCB. However, they have made constructive comments to provide more definitive evidence for your model, which we agree need to be addressed with new experiments where requested. The reviewers have also brought up that your study opens up new mechanistic and conceptual questions, for example how V0-V1 senses phagosomal pH, why ATG16L1 recruitment does not occur at lysosomes, and the purpose of ATG8/LC3 on the vesicles. While we expect you to conduct all of their suggested experiments regarding these points as we agree they are important to clarify your study and provide further evidence for the specificity of your model, we do not expect a mechanistic expansion. In addition, please completely address all reviewer points regarding rigour and reproducibility.

GENERAL GUIDELINES:

Text limits: Character count for an Article is < 40,000, not including spaces. Count includes title page, abstract, introduction, results, discussion, acknowledgments, and figure legends. Count does not include materials and methods, references, tables, or supplemental legends.

Figures: Articles may have up to 10 main text figures. Figures must be prepared according to the policies outlined in our Instructions to Authors, under Data Presentation, <https://jcb.rupress.org/site/misc/ifora.xhtml>. All figures in accepted manuscripts will be screened prior to publication.

*****IMPORTANT:** It is JCB policy that if requested, original data images must be made available. Failure to provide original images upon request will result in unavoidable delays in publication. Please ensure that you have access to all original microscopy and blot data images before submitting your revision. ***

Supplemental information: There are strict limits on the allowable amount of supplemental data. Articles may have up to 5 supplemental figures. Up to 10 supplemental videos or flash animations are allowed. A summary of all supplemental material should appear at the end of the Materials and methods section.

As you may know, the typical timeframe for revisions is three to four months. However, we at JCB realize that the implementation of social distancing and shelter in place measures that limit spread of COVID-19 also pose challenges to scientific researchers. Lab closures especially are preventing scientists from conducting experiments to further their research. Therefore, JCB has waived the revision time limit. We recommend that you reach out to the editors once your lab has reopened to decide on an appropriate time frame for resubmission. Please note that papers are generally considered through only one revision cycle, so any revised manuscript will likely be either accepted or rejected.

Thank you for this interesting contribution to Journal of Cell Biology. You can contact us at the journal office with any questions,

cellbio@rockefeller.edu or call (212) 327-8588.

Sincerely,

Tamotsu Yoshimori, PhD
Monitoring Editor

Andrea L. Marat, PhD
Senior Scientific Editor

Journal of Cell Biology

Reviewer #1 (Comments to the Authors (Required)):

Hooper et al. show that various CASM inducers promote Vo-V1 association, which is induced by the increase in ROS or the luminal pH. Vo-V1 then recruits ATG16L1 in a pump-independent manner, resulting in the conjugation of ATG8s to endolysosomal membranes. These results are convincing and well-presented. Although some of the findings may be predictable from previous studies (e.g., it is known that STING activation-induced CASM is mediated by SopF-sensitive V-ATPase-ATG16L1 interaction), this manuscript presents a full and comprehensive picture of CASM. This reviewer has some comments/suggestions to improve this manuscript.

1. Since many lysosomotropic reagents cause CASM, the involvement of the pump activity of V-ATPase should be carefully ruled out. Are the effects of BafA1 (100 nM) and SaliP (SaliP, 2.5 μ M) on lysosomal pH comparable? Can SaliP inhibit CQ-induced lysosomal swelling (as does BafA1) but not CQ-induced CASM?
2. In Fig. 4, the authors suggest that DPI and BafA1 inhibit the recruitment of V1A to OPZ-containing phagosomes. Don't these inhibitors affect the level of Vo complex? It would be ideal if the authors demonstrate the results of Vo staining as well.
3. To confirm the effect of SaliP on CASM, it would be helpful if the authors could show that LC3 is recruited to endolysosomes upon SaliP treatment in ATG13 KO cells but not in ATG16L1 K490A mutant cells.
4. Figs. 5 and 7: The authors mention "Vo-V1 association" in the text, but this is not appropriate. What the authors see is merely membrane association of V1A. Can the authors actually monitor Vo-V1 association?
5. Fig. 6. In the IP experiment, the authors see interaction between ATG16L1 and V1A. Is this direct interaction? Is Vo also included in the precipitants? Can ATG16L1 and V1A interact in the cytosol?

Reviewer #2 (Comments to the Authors (Required)):

Manuscript Nr: 202105112

Hooper et al., "V-ATPase is a universal regulator of LC3 associated phagocytosis and non-canonical autophagy"

The authors demonstrate that Rab7, LAMP1 and vesicular ATPase (vATPase) are recruited to phagosomes prior to LC3 during LC3 associated phagocytosis (LAP). Some acidification inhibitors promote, while others inhibit LAP. The authors suggest that NADPH oxidase inhibition with DPI increases phagosome acidification which then in turn leads to less vATPase recruitment to phagosomes. vATPase's V0-V1 association seems to directly interact with ATG16L1 to lipidate LC3 at phagosomal membranes. This interaction is selectively disrupted by the SopF protein of Salmonella which thereby escapes clearance after up-take. Therefore, the authors conclude that V0-V1 of vATPase recruits LC3 lipidation to endosomal membranes.

This is an interesting proposition but raises a number of questions, some of which should at least be discussed.

Major comments:

1. The authors argue that NADPH oxidase does not only promote LAP via acidification inhibition (line 238-240 and figure S2), but yet suggest that high pH in phagosomes is sensed for more vATPase recruitment. This should be clarified.
2. If V0-V1 complex formation determines ATG16L1 recruitment, why is LC3 not lipidated to lysosomal membranes? These vesicles should constitutively carry vATPase and serve as docking sites for the LC3 lipidation machinery. Furthermore, why is Bafilomycin A blocking activity of V0-V1 formation and LAP inhibition not seen in ATG13 deficient MCF10A cells? LC3-II seems unaffected or even slightly increased by brefeldin A treatment in figure 5b.
3. How is the pH sensed to recruit more V0-V1? Presumably, this could be Rab7 dependent for fusion with lysosomes, but how many LAP vesicles are actually Rab7 and LAMP1 positive? The author should provide a quantification along these lines

(Rab7+LC3+ and LAMP1+LC3+ phagosomes or all LC3+ phagosomes) for figures 1 and S1.

4. The authors primarily rely on inhibitors for their study. It would be interesting to overexpress V0 and V1 of vATPase in order to determine if this influences membrane lipidation by ATG16L1.

5. The authors argue that their combined DPI plus chloroquine treatment argues that NADPH oxidase produced ROS that just neutralizes the pH in phagosomes for LAP formation and that this can be replaced by chloroquine. However, at the same time phagocytosed cargo degradation is inhibited and the authors should at least discuss that this could lead to the accumulation of the during DPI treatment at lower rate forming LAP vesicles. Without cargo degradation these vesicles still engage zymosan dependent signaling, triggering LAP, and LC3 association with these might not only depend on the phagosomal pH. Along these lines a more standardized a linear phagosomal pH measurement than lysotracker intensity would also provide clarity.

Minor comments:

1. None of the Western blots is quantified. This should be done, and the figure legends should indicate how often the respective experiments were performed.

Reviewer #3 (Comments to the Authors (Required)):

The manuscript by Hooper et al. describes the role of the V-ATPase as a regulator of non-canonical autophagy processes, including LC3-associated phagocytosis. In addition to its role in canonical autophagy, the V-ATPase has traditionally been described to function in the last steps of non-canonical autophagy degradative processes to acidify the content of endolysosomal vesicles and facilitate cargo turnover. This traditional model implies first the conjugation of the ATG8/LC3 proteins to the membrane of an endolysosomal vesicle and then ATG8/LC3, possibly via facilitating fusion with lysosomes, promote the recruitment of the V-ATPase and acidic hydrolases to allow cargo degradation. Interestingly, the authors of this manuscript report that, instead the V-ATPase, independent of its function in acidification, first associates with the endolysosome membrane to then recruit the machinery involved in ATG8/LC3 conjugation allowing ATG8/LC3 association to endolysosomal membranes. Specifically, the assembled V-ATPase binds the autophagy protein ATG16L1 via its WD40 domain, involved in the conjugation of ATG8/LC3 during non-canonical autophagy processes. Finally, the authors describe that V-ATPase-ATG16L1 binding can be antagonized by the intracellular pathogen Salmonella via its protein SopF, connecting this molecular pathway with host responses against pathogens. Overall, this is a very interesting, well-written, and thought-provoking study, which presents experimental evidence in support of its main claims, but rigor could be higher in places. In turn, the authors need to address the below points to further support their study.

1. Noting that the authors show that the V-ATPase associates first with endolysosomal membranes prior to ATG8/LC3 conjugation it is inevitable to wonder what may be the main function of ATG8/LC3 in those vesicles? The authors speculate that the presence of ATG8/LC3 may help regulate the extent of lysosomal fusion, which in turn could lead to further recruitment of V-ATPase and more acidic vesicles proficient in cargo degradation. To help address this point the authors should include a pH-sensitive marker such as lysotracker, to be combined with the staining of LC3 and V-ATPase shown in Figure 1D. Presumably, the V-ATPase-positive LC3-positive vesicles may be more acidic than their V-ATPase-positive LC3-negative precursors. If this is not the case, the authors should then discuss potential alternative functions of conjugated ATG8/LC3. Importantly, the supporting data in Figure S1 testing timing of recruitment in entosis have not been statistically processed and rigor/reproducibility details are lacking, making it difficult to evaluate this important figure.

2. Many of the claims made by the authors rely on the usage of drugs, including to modulate the assembly of V-ATPase subunits with Bafilomycin A (Baf A). When Baf A is used as a single drug, like in Figure 2, the authors are encouraged to use alternative treatment options to improve rigor (see also below point 5). The same level of rigor should ideally be applied for other drugs as well, e.g DPI as the single pharmacological inhibitor of NADPH oxidase. More importantly, the authors are encouraged to use genetic interventions, e.g. overexpression of V-ATPase, to help support their claims, and ideally test epistasis where applicable (Figure 2). As is, the conclusion in Line 186/187 that data confirm requirement seems largely over-stated (similarly with the conclusion to Figure 3B/D in line 219/220).

3. An important test of cell-type claims made from Figure 3E-I: Can ROS/pH levels in BMDCs be modulated so that they behave as BMDMs in LAP assay and ATP6V1A recruitment?

4. From Figure 5, the authors state in Line 292/293 that the association of V0-V1 is sufficient for LC3 conjugation to endolysosomal membranes independent of V-ATPase pump activity; while this may be inferred from the proposed functions of Baf A vs. SaliP, this important mechanistic point is not directly tested and should be formally excluded. What does SaliP do in LAP-related assays in Figure 2 and 4?

5. Figure 6 tests the important point about ATG16L1 interaction with V-ATPase, however it includes no comments about rigor and reproducibility; this is key to evaluate these important data and the claims made from the results. Is the interaction direct or indirect?

6. It would be interesting to directly test whether the expression of Salmonella SopF compromises ATG16L1 binding to V-

ATPase, as speculated by the authors. The authors should include immunoprecipitation experiments to support results in Figure 7 (noting that Figure 7D is missing rigor and reproducibility statement).

7. While interesting, the claim made from Figure 8 about a functional role for non-canonical autophagy in Salmonella host responses is supported by only one line of evidence (dependence on ATG16L1 K490 position), and the authors are encouraged to develop the pathogen angle further (or potentially include in Figure 7). As is, it has not been convincingly shown that Salmonella infection induces a non-canonical autophagy response.

8. Besides the degradative processes involving the non-canonical functions of the autophagy machinery described in this article, there are secretory processes that similarly involve the conjugation of ATG8/LC3 to vesicles of endosomal origin, which therefore may also be considered non-canonical, and for which the relevance of V-ATPase yet remains to be tested. The authors should clarify this in the text and perhaps accordingly tone down the "universal regulatory role" of V-ATPase in non-canonical processes involving the autophagy machinery, also noting that their study has concentrated on non-canonical autophagy as a key cellular pathway in immunity, but have not investigated cancer, vision or neurodegeneration at this point.

9. Finally, while the model shown in Figure 9 is very useful, it would further help the reader if the authors could integrate/speculate on some missing important elements such as lysosomal fusion, and its potential contribution to the delivery of V-ATPase subunits to the endolysosomal membrane.

Textual points:

1. The manuscript would be improved for the non-expert reader if a more extensive description of the functions of ATG8/LC3 proteins was included in the introduction of the article.

2. The text writing needs to be corrected to describe the results in past tense and make sure that compound adjectives are hyphenated (including in title). Also, check for correct/consistent spelling of elements such as mTOR and H⁺.

3. Line 95 - the authors should refrain from using 'loaded' phrases like non-canonical autophagy is "best studied" in the context of LAP.

4. Line 110 - autophagosomes are not degraded, the inner membranes of autolysosomes (and the engulfed material, as stated) are.

5. Line 394 and 448 mean to refer to the figure with the model, ie Figure 9 not Figure 8.

6. Lines 527-530: Did all co-authors read and provide comments to the manuscript?

We thank the reviewers for their time and greatly appreciate their constructive feedback.

Reviewer #1 (Comments to the Authors (Required)):

Hooper et al. show that various CASM inducers promote Vo-V1 association, which is induced by the increase in ROS or the luminal pH. Vo-V1 then recruits ATG16L1 in a pump-independent manner, resulting in the conjugation of ATG8s to endolysosomal membranes. These results are convincing and well-presented. Although some of the findings may be predictable from previous studies (e.g., it is known that STING activation-induced CASM is mediated by SopF-sensitive V-ATPase-ATG16L1 interaction), this manuscript presents a full and comprehensive picture of CASM. This reviewer has some comments/suggestions to improve this manuscript.

We thank the reviewer for their positive comments and suggestions.

1. Since many lysosomotropic reagents cause CASM, the involvement of the pump activity of V-ATPase should be carefully ruled out. Are the effects of BafA1 (100 nM) and SaliP (SaliP, 2.5 μ M) on lysosomal pH comparable? Can SaliP inhibit CQ-induced lysosomal swelling (as does BafA1) but not CQ-induced CASM?

We agree with the reviewer that carefully excluding the involvement of v-ATPase pump activity is important. To strengthen our findings in this area, we have now confirmed that SaliP abrogates LysoTracker staining in a comparable manner to BafA1 (new Fig. 5 b), suggesting a similar effect on blocking V-ATPase pump activity. These findings are in agreement with previous publications (Kissing et al., 2015).

Despite their comparable effects on pump inhibition, SaliP and BafA1 yield opposing effects on CASM. We have included additional data to reinforce this observation, showing that, in contrast to BafA1, SaliP does not block monensin or entosis induced CASM (new Fig. 5 e-i). Indeed, all our data indicate that SaliP treatment is in fact sufficient to promote CASM, by forcing V0-V1 engagement and recruiting ATG16L1.

2. In Fig. 4, the authors suggest that DPI and BafA1 inhibit the recruitment of V1A to OPZ-containing phagosomes. Don't these inhibitors affect the level of Vo complex? It would be ideal if the authors demonstrate the results of Vo staining as well.

We agree with the reviewer that this would be an informative piece of data to include. Given that DPI treatment results in greater acidification of phagosomes, it seems likely that a functional and intact V-ATPase complex, including V0 sectors, must be present. However, we have been unable to confirm this experimentally due to limitations on reagents. We tested numerous anti-V0 antibodies, but were unable to find one that works satisfactorily for immunofluorescence. Indeed, this issue with antibody reagents against V-ATPase is a well-recognised limitation within the field, which unfortunately confounds this particular experiment.

3. To confirm the effect of SaliP on CASM, it would be helpful if the authors could show that LC3 is recruited to endolysosomes upon SaliP treatment in ATG13 KO cells but not in ATG16L1 K490A mutant cells.

Thanks for these suggestions. To address the question of ATG13 independence, we show that SaliP induces LC3 lipidation, which colocalises with LAMP1 staining (Fig. 5 c and d), in ATG13 null cells. These data are consistent with the notion that SaliP induces CASM, independent of upstream canonical autophagy machinery.

Assessment of a role for ATG16L1 K490 in this context is more complicated, because SaliP plays dual, parallel roles in promoting CASM but also blocking autophagic flux. As such, while we would expect CASM to be inhibited in ATG16L1 K490A cells we would expect SaliP to nevertheless increase lipidated LC3, due to the accumulation of basal autophagosomes, confounding this analysis.

As an alternative approach, we have included additional data to strengthen the conclusion that SaliP induces CASM, using entosis as an extra model. We have previously shown that entotic corpse vacuoles (ECVs) are single-membrane compartments which can be targeted for CASM during their physiological maturation (Florey et al., 2011), but also by a range of ionophores and lysosomotropic drugs (Florey et al., 2015; Jacquin et al., 2017). Using this system, we now show that SaliP significantly induces GFP-LC3 recruitment to LAMP1 positive ECVs (Fig. 5 e and f). Furthermore, we find that SaliP prolongs CASM associated with entotic cell death (Fig. 5 g-i). Together these data provide strong, complementary evidence that SaliP induces CASM.

4. Figs. 5 and 7: The authors mention "Vo-V1 association" in the text, but this is not appropriate. What the authors see is merely membrane association of V1A. Can the authors actually monitor Vo-V1 association?

To investigate the role of v-ATPase, we used a complementary approach including immunofluorescence, membrane fractionation and western blot techniques, similar to those used by other groups in this field (Fischer et al., 2020; Kissing et al., 2015; Liberman et al., 2014; McGuire and Forgac, 2018; Stransky and Forgac, 2015). Currently there are no reagents/techniques to monitor V0-V1 association more directly in cells, but reversible assembly of V0-V1 sectors is a well-established concept, typically monitored by V1A recruitment (and coincident acidification). As such, we feel our conclusions are in line with the current methods in the field.

5. Fig. 6. In the IP experiment, the authors see interaction between ATG16L1 and V1A. Is this direct interaction? Is Vo also included in the precipitants? Can ATG16L1 and V1A interact in the cytosol?

This is an important point. Presently, we are not able to say definitively that it is a direct interaction between ATG16L1 and V1A. We and others have found other V-ATPase subunits pulled down by ATG16L1 (V1B2, V1D, V0D1), suggesting it interacts with the whole V-ATPase complex (new Fig. S4 e) (Ulferts et al., 2021; Xu et al., 2022; Xu et al., 2019). We have been careful not state a direct interaction between ATG16L1 and V1A in the results or discussion, and further discussion on this point is included (lines 359-361).

We do not have any evidence to suggest ATG16L1 and V1A interact in the cytosol. Our model is that upon V0-V1 engagement at the membrane, a conformational change occurs in V-ATPase which allows ATG16L1 binding through the K490 region.

Reviewer #2 (Comments to the Authors (Required)):

Manuscript Nr: 202105112

Hooper et al., "V-ATPase is a universal regulator of LC3 associated phagocytosis and non-canonical autophagy"

The authors demonstrate that Rab7, LAMP1 and vesicular ATPase (vATPase) are recruited to phagosomes prior to LC3 during LC3 associated phagocytosis (LAP). Some acidification inhibitors promote, while others inhibit LAP. The authors suggest that NADPH oxidase inhibition with DPI increases phagosome acidification which then in turn leads to less vATPase recruitment to phagosomes. vATPase's V0-V1 association seems to directly interact with ATG16L1 to lipidate LC3 at phagosomal membranes. This interaction is selectively disrupted by the SopF protein of Salmonella which thereby escapes clearance after up-take. Therefore, the authors conclude that V0-V1 of vATPase recruits LC3 lipidation to endosomal membranes.

This is an interesting proposition but raises a number of questions, some of which should at least be discussed.

We thank the reviewer for their consideration of our manuscript and proposed model and helpful comments.

Major comments:

1. The authors argue that NADPH oxidase does not only promote LAP via acidification inhibition (line 238-240 and figure S2), but yet suggest that high pH in phagosomes is sensed for more vATPase recruitment. This should be clarified.

Thanks for this comment. The reviewer is correct in their interpretation of our data and model. We propose that in the case of LAP, NADPH oxidase, and ROS production, act to alkalinise the phagosome, and the cell in turn increases V-ATPase recruitment to counter this. We suggest it is this increased V-ATPase assembly which actually activates LAP. Indeed, all the inducers of CASM that we investigated converged upon v-ATPase assembly and recruitment of ATG16L1, suggesting this is a universal mechanism for non-canonical autophagy induction. We hope that Figure 10 clarifies this model

2. If V0-V1 complex formation determines ATG16L1 recruitment, why is LC3 not lipidated to lysosomal membranes? These vesicles should constitutively carry vATPase and serve as docking sites for the LC3 lipidation machinery. Furthermore, why is Bafilomycin A blocking activity of V0-V1 formation and LAP inhibition not seen in ATG13 deficient MCF10A cells? LC3-II seems unaffected or even slightly increased by brefeldin A treatment in figure 5b.

Why CASM is not observed on lysosomes basally is a very interesting point. We do not see constitutive activation of CASM on lysosomes, even though V-ATPase is present and active. Our thinking is that CASM is induced specifically on perturbed or neutralising lysosomes/lysosome-related organelles, driving an increase in V0-V1 association, and potentially a conformational change in V-ATPase associated with greater activity. In resting lysosomes, this state would perhaps not be activated. However, a greater structural understanding of the different states of the V-ATPase complex would be needed to test this

hypothesis, and understand this area better, which is something we hope to explore in the future.

We find the second part of the comment slightly confusing. The reviewer asks why BafA1 does not block CASM in an ATG13 null context, but we had not addressed this (indeed, such a result would run counter to our predictions). Also, we assume 'brefeldin A' is a typo, and that the final sentence is also focussed on BafA1 effects? We can provide some notes and additional data to clarify:

1) The dual effects of BafA1. As noted by the reviewer, BafA blocks V0-V1 association and is well known to inhibit LAP, and all other forms of CASM, thereby suppressing LC3 lipidation to single membranes. In parallel, BafA1 also blocks autophagic flux, thus 'trapping' and elevating LC3 lipidation on double membrane autophagosomes.

2) In WT cells, BafA1 yields these two, opposing effects in parallel. This manifests, on balance, as an induction of LC3 lipidation (from autophagosomes) under basal conditions. This is well known in the literature and we provide consistent data in the new Fig 5c (WT panel: BafA1 raises basal LC3-II).

3) In ATG13 null cells, where autophagosome formation is inhibited, BafA1 simply blocks CASM. This tends to have little effect on basal LC3 lipidation levels, and we provide consistent data in Fig 5c (ATG13 KO panel: BafA1 no longer raises basal LC3-II).

4) It is also important to clarify that BafA1 WOULD indeed be expected to block induced CASM. This is well established in the literature and we have added further data in Fig. 5f which confirms that bafA1 inhibits monensin induced LC3 lipidation to entotic corpse vacuoles.

5) Mechanistically, the key, novel point for us here is the clear contrast between SaliP and BafA1 in ATG13 KO conditions (Fig 5 c-f). SaliP drives CASM under basal conditions, while BafA1 does not, pointing to the pivotal role of enhanced V0-V1 engagement.

6) We also note that in unstimulated cells, BafA1 does not appear to reduce V0-V1 association by a large degree. We assume this is likely due to the kinetics and levels of V0-V1 engagement under basal conditions, and predict that overtime there would be more pronounced inhibition. The effect of BafA1 is much clearer when assaying its ability to block induced V0-V1 association, for instance in response to TRPML1 agonist, (see data below) demonstrating the negative effect of BafA1 on V0-V1 engagement.

We hope these notes, and additional data, help to clarify.

HeLa cells
 US = unstimulated
 A = AZD8055, 1uM (mTOR inhibitor)
 #1, #2 = TRPML1 agonists, 2uM
 BafA1 = Bafilomycin A1, 100nM

3. How is the pH sensed to recruit more V0-V1? Presumably, this could be Rab7 dependent for fusion with lysosomes, but how many LAP vesicles are actually Rab7 and LAMP1 positive? The author should provide a quantification along these lines (Rab7+LC3+ and LAMP1+LC3+ phagosomes or all LC3+ phagosomes) for figures 1 and S1.

How luminal pH changes are sensed, and how this is relayed to the V-ATPase, is an open and very interesting question. One suggestion is that a luminal facing part of a V-ATPase subunit undergoes a pH dependent conformation change, although this has not been tested directly. Exploring this issue would represent a major new project, and require significant developments in reagents and technology. We feel this falls beyond the scope of this paper.

With regards to Rab7 more specifically, our live imaging experiments showed that all LC3 phagosomes are Rab7 and LAMP1 positive, and that these markers are recruited prior to LC3. We have included additional quantification and statistical analysis in new Fig. 1 c.

4. The authors primarily rely on inhibitors for their study. It would be interesting to overexpress V0 and V1 of vATPase in order to determine if this influences membrane lipidation by ATG16L1.

While we understand the point raised, we are limited here by a lack of reagents available to study V-ATPase. V-ATPase is an essential complex, so KO or KD cells are difficult to make and use. There are also multiple isoforms of multiple subunits, making it unclear as to what redundancy there is in the system. A few studies have used V-ATPase subunit overexpression to interfere with the enzyme, but it is somewhat unclear as to what effect this has on V-ATPase function. In light of this uncertainty, we have steered clear of using over-expression systems to perturb V-ATPase function. We feel that, while limited in some respects, our pharmacological approach has proven to be very effective in this mechanistic study, because the striking dichotomy between SaliP and BafA enabled us to identify the importance of V0-V1 engagement.

5. The authors argue that their combined DPI plus chloroquine treatment argues that NADPH oxidase produced ROS that just neutralizes the pH in phagosomes for LAP formation and that this can be replaced by chloroquine. However, at the same time phagocytosed cargo degradation is inhibited and the authors should at least discuss that this could lead to the accumulation of the during DPI treatment at lower rate forming LAP vesicles. Without cargo degradation these vesicles still engage zymosan dependent signaling, triggering LAP, and LC3 association with these might not only depend on the phagosomal pH. Along these lines a more standardized a linear phagosomal pH measurement than lysotracker intensity would also provide clarity.

Thanks for this comment. As noted, CQ can reverse DPI mediated inhibition of LAP. To test whether this may relate to inhibition of cargo degradation, rather than sequestration of protons, we performed an alternative experiment where cells were co-treated with DPI and the cysteine/cathepsin inhibitors E64-d + pepstatinA. By monitoring GFP-LC3 recruitment, we found no effect of E64d+pep on DPI inhibition of LAP (see new Fig. S3). These data suggest that lack of cargo degradation does not impact LAP.

Minor comments:

1. None of the Western blots is quantified. This should be done, and the figure legends should indicate how often the respective experiments were performed.

We have now included western blot quantification and experimental repeat information as requested.

Reviewer #3 (Comments to the Authors (Required)):

The manuscript by Hooper et al. describes the role of the V-ATPase as a regulator of non-canonical autophagy processes, including LC3-associated phagocytosis. In addition to its role in canonical autophagy, the V-ATPase has traditionally been described to function in the last steps of non-canonical autophagy degradative processes to acidify the content of endolysosomal vesicles and facilitate cargo turnover. This traditional model implies first the conjugation of the ATG8/LC3 proteins to the membrane of an endolysosomal vesicle and then ATG8/LC3, possibly via facilitating fusion with lysosomes, promote the recruitment of the V-ATPase and acidic hydrolases to allow cargo degradation. Interestingly, the authors of this manuscript report that, instead the V-ATPase, independent of its function in acidification, first associates with the endolysosome membrane to then recruit the machinery involved in ATG8/LC3 conjugation allowing ATG8/LC3 association to endolysosomal membranes. Specifically, the assembled V-ATPase binds the autophagy protein ATG16L1 via its WD40 domain, involved in the conjugation of ATG8/LC3 during non-canonical autophagy processes. Finally, the authors describe that V-ATPase-ATG16L1 binding can be antagonized by the intracellular pathogen *Salmonella* via its protein SopF, connecting this molecular pathway with host responses against pathogens. Overall, this is a very interesting, well-written, and thought-provoking study, which presents experimental evidence in support of its main claims, but rigor could be higher in places. In turn, the authors need to address the below points to further support their study.

Thank you to the reviewer for their appreciation of the study.

1. Noting that the authors show that the V-ATPase associates first with endolysosomal membranes prior to ATG8/LC3 conjugation it is inevitable to wonder what may be the main function of ATG8/LC3 in those vesicles? The authors speculate that the presence of ATG8/LC3 may help regulate the extent of lysosomal fusion, which in turn could lead to further recruitment of V-ATPase and more acidic vesicles proficient in cargo degradation. To help address this point the authors should include a pH-sensitive marker such as lysotracker, to be combined with the staining of LC3 and V-ATPase shown in Figure 1D. Presumably, the V-ATPase-positive LC3-positive vesicles may be more acidic than their V-ATPase-positive LC3-negative precursors. If this is not the case, the authors should then discuss potential alternative functions of conjugated ATG8/LC3. Importantly, the supporting data in Figure S1 testing timing of recruitment in entosis have not been statistically processed and rigor/reproducibility details are lacking, making it difficult to evaluate this important figure.

Thanks for this comment. As requested, we have performed live confocal imaging to monitor LysoTracker Red (LTR) and GFP-hLC3A during LAP (new Fig. 1 f and g). Our data show that GFP-LC3 recruits just prior to substantial increases in LTR intensity. However, we do not have any evidence to suggest that LC3/CASM are required for subsequent acidification and at this stage the precise function of conjugated ATG8/LC3 at phagosomes remains unclear. We have expanded our discussion on this as suggested (lines 518-535).

With respect to Fig S1, we have included additional data on the timings of LAMP1-RFP and GFP-LC3 during entosis, which remain consistent with the conclusion that LAMP1 recruitment occurs prior to LC3 (new Fig. S1 c).

2. Many of the claims made by the authors rely on the usage of drugs, including to modulate the assembly of V-ATPase subunits with Bafilomycin A (Baf A). When Baf A is used as a single drug, like in Figure 2, the authors are encouraged to use alternative treatment options to improve rigor (see also below point 5). The same level of rigor should ideally be applied for other drugs as well, e.g. DPI as the single pharmacological inhibitor of NADPH oxidase. More importantly, the authors are encouraged to use genetic interventions, e.g. overexpression of V-ATPase, to help support their claims, and ideally test epistasis where applicable (Figure 2). As is, the conclusion in Line 186/187 that data confirm requirement seems largely over-stated (similarly with the conclusion to Figure 3B/D in line 219/220).

We have now expanded our pharmacological manipulation of V-ATPase and NOX2 to include 2 inhibitors (Bafilomycin A1 and Concanamycin A for V-ATPase, and DPI and GSK for NOX2). Using these inhibitors, we reinforce our findings that inhibition of V-ATPase or NOX2 leads to the reduction of LAP (new Fig. 2). We also note these findings are in line with a large body of published work which have used either pharmacological or genetic inhibition of NOX2 to block LAP (Gluschko et al., 2018; Huang et al., 2009; Martinez et al., 2015; Stempels et al., 2022).

As noted above (Reviewer 2, Point 4), the use of V-ATPase subunit overexpression to modulate mammalian V-ATPase, has only been used in a small number of studies which often do not link the observed phenotype to alterations in V-ATPase activity. Coupled with the complexity of multiple subunits and isoforms, it is unclear what redundancy exists within the system. Furthermore, over-expression studies are often confounded by levels of over-expression achieved, with too much or too little having differential effects. For these reasons, and in line with the majority of published work in this field, we have avoided this route of V-ATPase manipulation. We thank the reviewer for the suggestion to include additional inhibitors, which we feel has helped mitigate this limitation.

3. An important test of cell-type claims made from Figure 3E-I: Can ROS/pH levels in BMDCs be modulated so that they behave as BMDMs in LAP assay and ATP6V1A recruitment?

This is an interesting suggestion. We have added extra data to further interrogate the observed relationship between phagosomal ROS, VIA recruitment and LAP induction, in macrophage and dendritic cell types. Dendritic cells (BMDCs) have higher phagosomal ROS, phagosomal VIA and LAP than macrophage (BMDMs) (Fig. 3 e and i; Fig. 4 d and e). We show that treatment of dendritic cells with DPI, which reduces ROS, inhibits LAP and leads to reduced VIA recruitment, to levels similar to those seen in macrophage (new Fig. 4 e). Thus, we are able to modulate dendritic cells to behave more like macrophage in relation to LAP and VIA recruitment, which strengthens our proposed model.

4. From Figure 5, the authors state in Line 292/293 that the association of V0-V1 is sufficient for LC3 conjugation to endolysosomal membranes independent of V-ATPase pump activity; while this may be inferred from the proposed functions of Baf A vs. SaliP, this important

mechanistic point is not directly tested and should be formally excluded. What does SaliP do in LAP-related assays in Figure 2 and 4?

This is a very interesting point and related to our response to Reviewer 1-point 3. We have now included additional data confirming that BafA1 and SaliP inhibit V-ATPase pump activity, and lead to loss of LysoTracker staining, to a comparable degree (new Fig. 5 b). We also show that SaliP, but not BafA1, induces ATG13-independent LC3 lipidation, which colocalises with lysosomes (Fig. 5 c-d), suggestive of CASM activation. We go on to show that SaliP promotes GFP-LC3 recruitment to single-membrane, LAMP1-positive entotic corpse vacuoles (Fig. 5 e and f), similar to the effect of other activators of CASM (monensin), while BafA1 has no effect. We also show that during entotic cell killing, a physiologically relevant example of non-canonical autophagy (Florey et al., 2011), BafA1 inhibits GFP-LC3 recruitment to entotic vacuoles while SaliP does not (new Fig. 5 g). Indeed, SaliP actually prolongs the duration of GFP-LC3 on entotic vacuoles (new Fig. 5 h and i), likely due to the prolonged engagement of V0-V1.

Thus, these two distinct V-ATPase inhibitors, that both inhibit pump activity, have opposing effects on CASM, with SaliP promoting and BafA1 inhibiting. This is the rationale for testing the opposing effects of the inhibitors on V0-V1 association and its effect on CASM, explored in the following figures.

5. Figure 6 tests the important point about ATG16L1 interaction with V-ATPase, however it includes no comments about rigor and reproducibility; this is key to evaluate these important data and the claims made from the results. Is the interaction direct or indirect?

Thanks for this comment. We have now included quantification of ATG16L1 pull-downs (new Fig. S4), and information on number of repeats in figure legends, to provide a more complete and robust data set.

Currently it is unclear whether the interaction between ATG16L1 and V-ATPase is direct or not, or what subunit of V-ATPase may support the interaction. We have added extra discussion points on this matter (please also see response to Reviewer 1-point 6).

6. It would be interesting to directly test whether the expression of Salmonella SopF compromises ATG16L1 binding to V-ATPase, as speculated by the authors. The authors should include immunoprecipitation experiments to support results in Figure 7 (noting that Figure 7D is missing rigor and reproducibility statement).

The inhibitory effect of SopF on the interaction between ATG16L1 and V-ATPase has previously been shown by two independent groups (Ulferts et al., 2021; Xu et al., 2022; Xu et al., 2019), which are cited in the manuscript.

We have added a reproducibility statement for new Fig. 8 d.

7. While interesting, the claim made from Figure 8 about a functional role for non-canonical autophagy in Salmonella host responses is supported by only one line of evidence (dependence on ATG16L1 K490 position), and the authors are encouraged to develop the pathogen angle further (or potentially include in Figure 7). As is, it has not been convincingly shown that Salmonella infection induces a non-canonical autophagy response.

We agree with the reviewer that exploring a functional role for non-canonical autophagy in Salmonella infection is an interesting angle to follow. In the paper, we use dependence on ATG16L1 K490A to link Salmonella infection to CASM; this point mutation is being specifically associated with non-canonical autophagy by a collection of studies. We feel these data make an important addition to the current paper, leading us to SopF and manipulation of the v-ATPase-ATG16L1 axis. To develop the link between Salmonella and non-canonical autophagy more completely, other defining features of CASM would include: a) the involvement of single membranes, rather than double membrane autophagosomes, and b) dependence on core ATG machinery (e.g. ATG5,7), combined with independence of upstream regulators (e.g. ULK1). In the present study, we have not undertaken electron microscopy, or further genetic analyses, largely because these findings have actually already been published elsewhere:

1) Salmonella are targeted by LC3 recruitment to single-membrane phagosomes (Masud et al., 2019).

2) Furthermore, this LC3 recruitment is independent of ATG13, while being dependent on ATG5, Rubicon and NADPH oxidase, which are some hallmark features of LAP.

3) Similarly, earlier work showed that LC3 recruitment to Salmonella occurs in the absence of ATG9, PI3K and FIP200 (Kageyama et al., 2011), which would also be consistent with CASM/LAP. The conclusions from this study were that Salmonella containing single-membrane compartments were directly targeted for LC3 lipidation, in parallel to some xenophagy/autophagosome targeting.

By considering these published findings, alongside our data on K490A, we feel there is a compelling case for CASM function during Salmonella infection, as outlined in our discussion. Exploring this area in greater depth would extend beyond the focus of the current work, but may be of interest for future studies.

8. Besides the degradative processes involving the non-canonical functions of the autophagy machinery described in this article, there are secretory processes that similarly involve the conjugation of ATG8/LC3 to vesicles of endosomal origin, which therefore may also be considered non-canonical, and for which the relevance of V-ATPase yet remains to be tested. The authors should clarify this in the text and perhaps accordingly tone down the "universal regulatory role" of V-ATPase in non-canonical processes involving the autophagy machinery, also noting that their study has concentrated on non-canonical autophagy as a key cellular pathway in immunity, but have not investigated cancer, vision or neurodegeneration at this point.

We agree with the reviewer that the functions of non-canonical autophagy are not fully understood, and have expanded on the potential roles of CASM in the discussion section accordingly.

We appreciate the point that certain non-canonical autophagy processes exist that have not been tested here. To a certain extent, this is complicated by the diversity of processes now known to harness subsets of the ATG machinery, and by the sometimes confusing and conflicting terminologies used to describe them. Taking the secretory example given, it is not yet clear whether the machinery and mechanisms involved are the same, or overlapping, with processes such as LAP.

We do feel confident that with respect to CASM, we have investigated multiple, diverse stimuli, including LAP, entosis, STING activation, TRPML1 agonists and lysosomotropic drugs, which together with recent work on influenza infection (Ulferts et al., 2021), point to a clear unifying mechanism of CASM involving V-ATPase. Examples of non-canonical autophagy involved in cancer and vision are mediated by LAP, and would be expected to involve the V-ATPase mechanisms as described in our study. Neurodegeneration involves LANDO (LC3 associated endocytosis), and is similarly dependent on Rubicon and ROS, so would therefore likely feed into the same V-ATPase mechanism. We have expanded the discussion section to reflect on this proposal.

9. Finally, while the model shown in Figure 9 is very useful, it would further help the reader if the authors could integrate/speculate on some missing important elements such as lysosomal fusion, and its potential contribution to the delivery of V-ATPase subunits to the endolysosomal membrane.

We thank the reviewer for this point and have included additional discussion points around the function of ATG8 proteins at endolysosomal membranes (lines 518-535).

Textual points:

1. The manuscript would be improved for the non-expert reader if a more extensive description of the functions of ATG8/LC3 proteins was included in the introduction of the article.

We have expanded on the known roles of ATG8 proteins in the introduction and discussion sections (lines 74-77).

2. The text writing needs to be corrected to describe the results in past tense and make sure that compound adjectives are hyphenated (including in title). Also, check for correct/consistent spelling of elements such as mTOR and H+.

We have addressed these issues.

3. Line 95 - the authors should refrain from using 'loaded' phrases like non-canonical autophagy is "best studied" in the context of LAP.

When referring to “non-canonical autophagy” as processes that are associated with CASM, LAP is clearly the most studied example when looking at number of publications. We have altered the text to make it clear this comment refers to most “commonly studied”.

4. Line 110 - autophagosomes are not degraded, the inner membranes of autolysosomes (and the engulfed material, as stated) are.

We have made this text change.

5. Line 394 and 448 mean to refer to the figure with the model, ie Figure 9 not Figure 8.

Thank you for pointing out this error. We have now corrected it.

6. Lines 527-530: Did all co-authors read and provide comments to the manuscript?

We can confirm that the manuscript was sent to all co-authors for comments.

References

- Fischer, T.D., C. Wang, B.S. Padman, M. Lazarou, and R.J. Youle. 2020. STING induces LC3B lipidation onto single-membrane vesicles via the V-ATPase and ATG16L1-WD40 domain. *J Cell Biol.* 219.
- Florey, O., N. Gammoh, S.E. Kim, X. Jiang, and M. Overholtzer. 2015. V-ATPase and osmotic imbalances activate endolysosomal LC3 lipidation. *Autophagy.* 11:88-99.
- Florey, O., S.E. Kim, C.P. Sandoval, C.M. Haynes, and M. Overholtzer. 2011. Autophagy machinery mediates macroendocytic processing and entotic cell death by targeting single membranes. *Nat Cell Biol.* 13:1335-1343.
- Gluschko, A., M. Herb, K. Wiegmann, O. Krut, W.F. Neiss, O. Utermöhlen, M. Krönke, and M. Schramm. 2018. The $\beta(2)$ Integrin Mac-1 Induces Protective LC3-Associated Phagocytosis of *Listeria monocytogenes*. *Cell host & microbe.* 23:324-337.e325.
- Huang, J., V. Canadien, G.Y. Lam, B.E. Steinberg, M.C. Dinauer, M.A. Magalhaes, M. Glogauer, S. Grinstein, and J.H. Brummell. 2009. Activation of antibacterial autophagy by NADPH oxidases. *Proceedings of the National Academy of Sciences of the United States of America.* 106:6226-6231.
- Jacquin, E., S. Leclerc-Mercier, C. Judon, E. Blanchard, S. Fraitag, and O. Florey. 2017. Pharmacological modulators of autophagy activate a parallel noncanonical pathway driving unconventional LC3 lipidation. *Autophagy.* 13:854-867.
- Kageyama, S., H. Omori, T. Saitoh, T. Sone, J.L. Guan, S. Akira, F. Imamoto, T. Noda, and T. Yoshimori. 2011. The LC3 recruitment mechanism is separate from Atg9L1-dependent membrane formation in the autophagic response against *Salmonella*. *Mol Biol Cell.* 22:2290-2300.
- Kissing, S., C. Hermsen, U. Repnik, C.K. Nasset, K. von Bargen, G. Griffiths, A. Ichihara, B.S. Lee, M. Schwake, J. De Brabander, A. Haas, and P. Saftig. 2015. Vacuolar ATPase in phagosome-lysosome fusion. *J Biol Chem.* 290:14166-14180.
- Liberman, R., S. Bond, M.G. Shainheit, M.J. Stadecker, and M. Forgac. 2014. Regulated assembly of vacuolar ATPase is increased during cluster disruption-induced maturation of dendritic cells through a phosphatidylinositol 3-kinase/mTOR-dependent pathway. *J Biol Chem.* 289:1355-1363.
- Martinez, J., R.K. Malireddi, Q. Lu, L.D. Cunha, S. Pelletier, S. Gingras, R. Orchard, J.L. Guan, H. Tan, J. Peng, T.D. Kanneganti, H.W. Virgin, and D.R. Green. 2015. Molecular characterization of LC3-associated phagocytosis reveals distinct roles for Rubicon, NOX2 and autophagy proteins. *Nat Cell Biol.* 17:893-906.
- Masud, S., T.K. Prajsnar, V. Torraca, G.E.M. Lamers, M. Benning, M. Van Der Vaart, and A.H. Meijer. 2019. Macrophages target *Salmonella* by Lc3-associated phagocytosis in a systemic infection model. *Autophagy.* 15:796-812.
- McGuire, C.M., and M. Forgac. 2018. Glucose starvation increases V-ATPase assembly and activity in mammalian cells through AMP kinase and phosphatidylinositide 3-kinase/Akt signaling. *J Biol Chem.* 293:9113-9123.

- Stempels, F.C., M.H. Janssens, M. Ter Beest, R.J. Mesman, N.H. Revelo, M. Ioannidis, and G. van den Bogaart. 2022. Novel and conventional inhibitors of canonical autophagy differently affect LC3-associated phagocytosis. *FEBS letters*.
- Stransky, L.A., and M. Forgac. 2015. Amino Acid Availability Modulates Vacuolar H⁺-ATPase Assembly. *J Biol Chem*. 290:27360-27369.
- Ulferts, R., E. Marcassa, L. Timimi, L.C. Lee, A. Daley, B. Montaner, S.D. Turner, O. Florey, J.K. Baillie, and R. Beale. 2021. Subtractive CRISPR screen identifies the ATG16L1/vacuolar ATPase axis as required for non-canonical LC3 lipidation. *Cell reports*. 37:109899.
- Xu, Y., S. Cheng, H. Zeng, P. Zhou, Y. Ma, L. Li, X. Liu, F. Shao, and J. Ding. 2022. ARF GTPases activate Salmonella effector SopF to ADP-ribosylate host V-ATPase and inhibit endomembrane damage-induced autophagy. *Nat Struct Mol Biol*. 29:67-77.
- Xu, Y., P. Zhou, S. Cheng, Q. Lu, K. Nowak, A.K. Hopp, L. Li, X. Shi, Z. Zhou, W. Gao, D. Li, H. He, X. Liu, J. Ding, M.O. Hottiger, and F. Shao. 2019. A Bacterial Effector Reveals the V-ATPase-ATG16L1 Axis that Initiates Xenophagy. *Cell*. 178:552-566.e520.

March 7, 2022

RE: JCB Manuscript #202105112R

Dr. Oliver Florey
Babraham Institute
Signalling Programme The Babraham Institute
Cambridge CB22 3AT
United Kingdom

Dear Dr. Florey:

Thank you for submitting your revised manuscript entitled "V-ATPase is a universal regulator of LC3-associated phagocytosis and non-canonical autophagy". We would be happy to publish your paper in JCB pending final revisions necessary to meet our formatting guidelines (see details below). In your final revision, please be sure to address the reviewers' final concerns with appropriate text edits. We agree the limitations of the study need to be made clear including in the abstract, however including this in the title is not consistent with journal requirements.

A. MANUSCRIPT ORGANIZATION AND FORMATTING:

- 1) Text limits: Character count for Articles is < 40,000, not including spaces. Count includes abstract, introduction, results, discussion, and acknowledgments. Count does not include title page, figure legends, materials and methods, references, tables, or supplemental legends.
- 2) Figures limits: Articles may have up to 10 main text figures.
- 3) * Figure formatting: Scale bars must be present on all microscopy images, including inset magnifications (you may alternatively indicate the diameter of the inset). Molecular weight or nucleic acid size markers must be included on all gel electrophoresis. Please avoid the use of red/green in graphs without additional distinguishing features (e.g. symbols). *
- 4) Statistical analysis: Error bars on graphic representations of numerical data must be clearly described in the figure legend. The number of independent data points (n) represented in a graph must be indicated in the legend. Statistical methods should be explained in full in the materials and methods. For figures presenting pooled data the statistical measure should be defined in the figure legends. Please also be sure to indicate the statistical tests used in each of your experiments (either in the figure legend itself or in a separate methods section) as well as the parameters of the test (for example, if you ran a t-test, please indicate if it was one- or two-sided, etc.). Also, if you used parametric tests, please indicate if the data distribution was tested for normality (and if so, how). If not, you must state something to the effect that "Data distribution was assumed to be normal but this was not formally tested."
- 5) Abstract and title: The abstract should be no longer than 160 words and should communicate the significance of the paper for a general audience. The title should be less than 100 characters including spaces. Make the title concise but accessible to a general readership.
- 6) Materials and methods: Should be comprehensive and not simply reference a previous publication for details on how an experiment was performed. Please provide full descriptions in the text for readers who may not have access to referenced manuscripts.
- 7) Please be sure to provide the sequences for all of your primers/oligos and RNAi constructs in the materials and methods. You must also indicate in the methods the source, species, and catalog numbers (where appropriate) for all of your antibodies. Please also indicate the acquisition and quantification methods for immunoblotting/western blots.
- 8) Microscope image acquisition: The following information must be provided about the acquisition and processing of images:
 - a. Make and model of microscope
 - b. Type, magnification, and numerical aperture of the objective lenses
 - c. Temperature
 - d. Imaging medium

- e. Fluorochromes
- f. Camera make and model
- g. Acquisition software
- h. Any software used for image processing subsequent to data acquisition. Please include details and types of operations involved (e.g., type of deconvolution, 3D reconstitutions, surface or volume rendering, gamma adjustments, etc.).

10) Supplemental materials: There are strict limits on the allowable amount of supplemental data. Articles may have up to 5 supplemental figures. Please also note that tables, like figures, should be provided as individual, editable files. A summary of all supplemental material should appear at the end of the Materials and methods section.

13) ORCID IDs: ORCID IDs are unique identifiers allowing researchers to create a record of their various scholarly contributions in a single place. At resubmission of your final files, please consider providing an ORCID ID for as many contributing authors as possible.

Please note that JCB now requires authors to submit Source Data used to generate figures containing gels and Western blots with all revised manuscripts. This Source Data consists of fully uncropped and unprocessed images for each gel/blot displayed in the main and supplemental figures. Since your paper includes cropped gel and/or blot images, please be sure to provide one Source Data file for each figure that contains gels and/or blots along with your revised manuscript files. File names for Source Data figures should be alphanumeric without any spaces or special characters (i.e., SourceDataF#, where F# refers to the associated main figure number or SourceDataFS# for those associated with Supplementary figures). The lanes of the gels/blots should be labeled as they are in the associated figure, the place where cropping was applied should be marked (with a box), and molecular weight/size standards should be labeled wherever possible.

B. FINAL FILES:

**The license to publish form must be signed before your manuscript can be sent to production. A link to the electronic license to

publish form will be sent to the corresponding author only. Please take a moment to check your funder requirements before choosing the appropriate license.**

Thank you for this interesting contribution, we look forward to publishing your paper in Journal of Cell Biology.

Sincerely,

Tamotsu Yoshimori, PhD
Monitoring Editor

Andrea L. Marat, PhD
Senior Scientific Editor

Journal of Cell Biology

Reviewer #1 (Comments to the Authors (Required)):

The authors have addressed the criticisms raised by this reviewer.

Reviewer #2 (Comments to the Authors (Required)):

Manuscript Nr: 202105112R

Hooper et al., "V-ATPase is a universal regulator of LC3 associated phagocytosis and non-canonical autophagy"

The authors demonstrate that Rab7, LAMP1 and vesicular ATPase (vATPase) are recruited to phagosomes prior to LC3 during LC3 associated phagocytosis (LAP). Some acidification inhibitors promote, while others inhibit LAP. The authors suggest that NADPH oxidase inhibition with DPI increases phagosome acidification which then in turn leads to less vATPase recruitment to phagosomes. vATPase's V0-V1 association seems to directly interact with ATG16L1 to lipidate LC3 at phagosomal membranes. This interaction is selectively disrupted by the SopF protein of Salmonella which thereby escapes clearance after up-take. Therefore, the authors conclude that V0-V1 of vATPase recruits LC3 lipidation to endosomal membranes.

In the revised manuscript the authors have addressed most of my concerns by quantifying data sets and expanding their discussion. However, the nearly exclusive reliance on pharmacological inhibition to manipulate V-ATPase and its association with ATG16L1 weakens the interpretation of the presented data. If the authors do not want to attempt V-ATPase subunit overexpression studies, the pharmacological inhibition of V-ATPase should be more prominently stated in title and abstract. The inhibitors should be mentioned in the abstract and "pharmacological inhibition of V-ATPase" should be indicated in the title.

Reviewer #3 (Comments to the Authors (Required)):

The authors have for the most part addressed this reviewer's concerns by providing several new pieces of experimental evidence and discussion points. The revised manuscript is recommended for publication with the following edits:

1. This reviewer understands the authors' reasoning regarding use of additional drugs over genetic alterations of the V-ATPase. However, this point remains a potential limitation to the interpretation of the study, which should be better reflected textually, including in the abstract.
2. The authors are encouraged to include the name of the new drug interventions used in the revised manuscript in their model

depicted in Figure 10.

3. Moreover, Figure 10 could further benefit from numbering to indicate the chronology of the steps of the model, with vesicle acidification happening after ATG8 proteins incorporation, as determined by the authors in their revised Figure 1.

4. Finally, please edit to write all Results in past tense (eg., correct "Firstly, while DPI and BafA1 both inhibit LAP, they yield different effects on phagosomal pH (Fig. S2)." - 'yield' and 'inhibit' need to be written in past tense).

5. Statistics seems to be missing in Fig 4e on DPI/BMDM.

Reviewer #1 (Comments to the Authors (Required)):

The authors have addressed the criticisms raised by this reviewer.

Thank you for all your helpful comments.

Reviewer #2 (Comments to the Authors (Required)):

Manuscript Nr: 202105112R

Hooper et al., "V-ATPase is a universal regulator of LC3 associated phagocytosis and non-canonical autophagy"

The authors demonstrate that Rab7, LAMP1 and vesicular ATPase (vATPase) are recruited to phagosomes prior to LC3 during LC3 associated phagocytosis (LAP). Some acidification inhibitors promote, while others inhibit LAP. The authors suggest that NADPH oxidase inhibition with DPI increases phagosome acidification which then in turn leads to less vATPase recruitment to phagosomes. vATPase's V0-V1 association seems to directly interact with ATG16L1 to lipidate LC3 at phagosomal membranes. This interaction is selectively disrupted by the SopF protein of Salmonella which thereby escapes clearance after up-take. Therefore, the authors conclude that V0-V1 of vATPase recruits LC3 lipidation to endosomal membranes.

In the revised manuscript the authors have addressed most of my concerns by quantifying data sets and expanding their discussion. However, the nearly exclusive reliance on pharmacological inhibition to manipulate V-ATPase and its association with ATG16L1 weakens the interpretation of the presented data. If the authors do not want to attempt V-ATPase subunit overexpression studies, the pharmacological inhibition of V-ATPase should be more prominently stated in title and abstract. The inhibitors should be mentioned in the abstract and "pharmacological inhibition of V-ATPase" should be indicated in the title.

Thank you for your helpful comments. We have now included additional text in the abstract and results to highlight the limitations in using pharmacological means to inhibit V-ATPase.

Reviewer #3 (Comments to the Authors (Required)):

The authors have for the most part addressed this reviewer's concerns by providing several new pieces of experimental evidence and discussion points. The revised manuscript is recommended for publication with the following edits:

Thank you for your helpful comments.

1. This reviewer understands the authors' reasoning regarding use of additional drugs over genetic alterations of the V-ATPase. However, this point remains a potential limitation to the interpretation of the study, which should be better reflected textually, including in the abstract.

As above, we have included additional text to address this issue.

2. The authors are encouraged to include the name of the new drug interventions used in the revised manuscript in their model depicted in Figure 10.

We have now included this in Figure 10.

3. Moreover, Figure 10 could further benefit from numbering to indicate the chronology of the steps of the model, with vesicle acidification happening after ATG8 proteins incorporation, as determined by the authors in their revised Figure 1.

We have included numbered steps into Figure 10 and the Figure Legend.

4. Finally, please edit to write all Results in past tense (eg., correct "Firstly, while DPI and BafA1 both inhibit LAP, they yield different effects on phagosomal pH (Fig. S2)." - 'yield' and 'inhibit' need to be written in past tense).

We have adjusted all the results to be written in past tense.

5. Statistics seems to be missing in Fig 4e on DPI/BMDM.

This statistic has been added.